# Trim33 masks a non-transcriptional function of E2f4 in replication fork progression

Vanessa Rousseau[1,2,6], Elias Einig [1], Chao Jin [1], Julia Horn[2,7], Mathias Riebold[3], Tanja Poth[4], Mohamed-Ali Jarboui [5], Michael Flentje[2] & Nikita Popov [1] ✉

Replicative stress promotes genomic instability and tumorigenesis but also presents an effective therapeutic endpoint, rationalizing detailed analysis of pathways that control DNA replication. We show here that the transcription factor E2f4 recruits the DNA helicase Recql to facilitate progression of DNA replication forks upon drug- or oncogene-induced replicative stress. In unperturbed cells, the Trim33 ubiquitin ligase targets E2f4 for degradation, limiting its genomic binding and interactions with Recql. Replicative stress blunts Trim33-dependent ubiquitination of E2f4, which stimulates transient Recql recruitment to chromatin and facilitates recovery of DNA synthesis. In contrast, deletion of Trim33 induces chronic genome-wide recruitment of Recql and strongly accelerates DNA replication under stress, compromising checkpoint signaling and DNA repair. Depletion of Trim33 in Myc-overexpressing cells leads to accumulation of replication-associated DNA damage and delays Myc-driven tumorigenesis. We propose that the Trim33-E2f4-Recql axis controls progression of DNA replication forks along transcriptionally active chromatin to maintain genome integrity.

Oncogene-induced replicative stress is a hallmark of cancer[1]. Different oncogenes transform cells via distinct mechanisms but commonly lead to accelerated passage through the cell cycle, deregulation of transcription, and DNA replication[2]. Deregulated DNA replication is a major source of endogenous DNA damage, which underlies tumor heterogeneity and the development of aggressive, therapy-resistant tumors[2,3]. Stalling or slowing of DNA replication forks both under stress conditions (e.g., nucleotide depletion or exposure to therapeutic genotoxins) and during the normal cell cycle, activates DDR signaling, which promotes recruitment of factors that stabilize replication forks, repair DNA and mediates restart of DNA synthesis[2–4]. Aberrant or accelerated progression of replication forks, for example, upon inhibition of Parp1, or bypass of DNA lesions by translesion

synthesis DNA polymerases, can lead to the accumulation of DNA damage and compromise genomic integrity[5,6].

The deregulated activity of RNA Polymerase II (RNAPII) is a major factor underlying replicative stress in tumor cells[7,8]. Ectopic RNAPII activation can cause DNA torsional stress, excessive formation of DNA-RNA hybrids (R-loops), and collisions of RNAPII and DNA polymerase complexes (transcription-replication conflicts, TRCs). Accordingly, oncogenes such as Myc and Ras, that deregulate RNAPII also perturb DNA replication[9,10].

The Myc oncogenic transcription factor is a key genome-wide regulator of RNAPII that recruits multiple factors to control transcriptional pausing, processive elongation, RNA maturation, and export[11,12]. Myc controls the expression of hundreds to thousands of

[1]Department of Medical Oncology and Pulmonology, University Hospital Tübingen, Otfried-Müller-Str 14, 72076 Tübingen, Germany. [2]Department of Radiation Oncology, University Hospital Würzburg, Josef-Schneider-Str 2, 97080 Würzburg, Germany. [3]Department of Gastroenterology, Gastrointestinal Oncology, Hepatology, Infectiology, and Geriatry, University Hospital Tübingen, Otfried-Müller-Str 12, 72076 Tübingen, Germany. [4]Center for Model System and Comparative Pathology, Institute of Pathology, University Hospital Heidelberg, Im Neuenheimer Feld 224, 69120 Heidelberg, Germany. [5]Core Facility for Medical Bioanalytics, Proteomics Platform Tübingen (PxP), Institute for Ophthalmic Research, University of Tübingen, Elfriede-Aulhorn-Str 7, 72076 Tübingen, Germany. [6]Present address: Interfaculty Institute for Biochemistry, University Hospital Tübingen, Auf der Morgenstelle 34, 72076 Tübingen, Germany. [7]Present address: Wakenitzmauer 3, 23552 Lübeck, Germany. ✉e-mail: nikita.popov@med.uni-tuebingen.de

genes and can thereby regulate many aspects of tumor development, including cell growth, proliferation, and metabolism. Beyond the function of individual genes, the genome-wide Myc-dependent effects on RNAPII can have a profound impact on cell fate. For example, pervasive activation of RNAPII activity leads to DNA damage and can underlie Myc-induced apoptosis[13]. In line with this idea, the development of Myc-overexpressing tumors often requires additional genetic alterations that allow tumor cell survival and proliferation under stress and high levels of DNA damage[13,14]. In the presence of such cooperating events, activation of Myc potently induces tumor formation in multiple tissues, including the lung, pancreas, skin, and the liver[11,14]. Consequently, cells overexpressing Myc are characterized by high levels of DNA damage and can be sensitized to inhibition of DNA damage response kinases ATR and ATM[15–17].

Myc function is stringently regulated by the ubiquitin-proteasome system. Several ubiquitin ligases target Myc for degradation but also regulate its function in a non-proteolytic manner[18]. We have previously identified the Trim33 ubiquitin ligase in an shRNA screen for genes that regulate cell survival upon Myc activation[19]. Recent studies implicated Trim33 in controlling transcription, response to DNA damage, and apoptosis[20–22]. Since Myc induces replicative stress and DNA damage, which can contribute to Myc-induced apoptosis, we sought to investigate the role of Trim33 in oncogene- and drug-induced replicative stress.

We show that Trim33 regulates DNA replication limiting Myc- and drug-induced replicative stress. Mechanistically, Trim33 targets for degradation of transcription factor E2f4 and thereby restricts E2f4 interactions with chromatin and with the Recql DNA helicase. Replication checkpoint signaling transiently inhibits Trim33-dependent regulation to allow E2f4-dependent recruitment of Recql to chromatin, which facilitates the progression of DNA replication forks. Genetic deletion of Trim33 induces constitutive recruitment of Recql to chromatin, accelerates replication forks, and compromises checkpoint signaling and DNA repair under replicative stress. Consequently, loss of Trim33 leads to the accumulation of DNA damage and delays the development of Myc-driven tumors.

## Results

### Deletion of Trim33 abates replicative stress

We used CRISPR to abolish the expression of Trim33 in a murine liver cancer cell line (p19/Nras), derived from an autochthonous tumor induced by Nras expression in the p19/Arf-null mice, which provides a tractable genetic model for in vitro and in vivo analysis (Fig. 1a)[23]. Knockout of Trim33 did not impact the proliferation rate and cell cycle distribution of unchallenged cells (Supplementary Fig. 1a, b). Transcriptional profiling in Trim33-WT and two Trim33-KO cell lines cells using RNA-seq identified more than 4000 commonly deregulated genes (p adj <0.01); of these, transcripts associated with DNA replication (KEGG Pathway) were the top enriched group (Fig. 1b). We, therefore, compared replication fork progression in control and Myc-overexpressing Trim33-WT and Trim33-KO cells using the DNA fiber assays. Deletion of Trim33 did not significantly affect replication fork progression in control cells, but rescued Myc-induced fork slowing (Fig. 1c). Similar results were obtained using shRNA-mediated depletion of Trim33 in cells derived from tumors, induced by Nras alone versus Nras and Myc—p19/Nras and p19/Nras/Myc[23] (Supplementary Fig. 1c, d). Analogously, CRISPR-mediated deletion of TRIM33 in U2OS cells with inducible expression of Myc[24] rescued Myc-induced fork slowing (Supplementary Fig. 1e, f), suggesting that Trim33 controls DNA replication in different cellular contexts. Acute depletion of Trim33 using shRNAs in U2OS cells also reverted slowing of replication forks, induced by high Myc levels (Supplementary Fig. 1g, h), indicating that the effect of Trim33 depletion on replication forks is direct and not caused by long-term changes in the absence of Trim33. Knockout of Trim33 reverted Myc-induced increase in S4/8-phosphorylated

Rpa2 (a marker of collapsed replication forks) and S824-phosphorylated Kap1 (a marker of ATM activation) (Fig. 1d; Supplementary Fig. 1i), indicating that depletion of Trim33 suppresses Myc-induced replicative stress and the accompanying activation of DDR signaling.

Deletion of Trim33 increased fork rates and stimulated total cellular DNA synthesis following release from hydroxyurea (HU) treatment (Fig. 1e, f), showing that Trim33 function is not limited to Myc-induced replicative stress. Similar results were obtained during recovery from aphidicolin and etoposide, which also induce fork stalling (Supplementary Fig. 1j), suggesting that Trim33 specifically restricts DNA replication during the restart of stalled replication forks under stress conditions. As in Myc-overexpressing cells, deletion of Trim33 diminished levels of pS4/8-Rpa2, pS824-Kap1, and pS317-Chk1 upon HU treatment (Fig. 1g). In line with reduced levels of replicative stress, entry into mitosis following release from HU was accelerated in Trim33-KO cells compared to Trim33-WT cells (Fig. 1h). In contrast, Trim33 loss did not strongly affect the onset of DNA replication after release from serum deprivation, which synchronizes cells in G1 (Fig. 1i). We concluded that loss of Trim33 facilitates DNA replication and compromises checkpoint signaling specifically under drug- or oncogene-induced replicative stress.

### Accelerated DNA replication in Trim33-KO cells requires transcription factor E2f4

Although endogenous Myc protein and mRNA were downregulated by loss of Trim33, the exogenous Myc protein levels in most experiments were not strongly affected (Supplementary Fig. 2a; Fig. 1d; Supplementary Fig. 1c, g, i). Furthermore, Myc stability was unaffected by the deletion of Trim33, and Myc ubiquitination was not enhanced by overexpression of Trim33 (Supplementary Fig. 2b, c), indicating that Trim33 regulates response to replicative stress via other factors. Analysis of Trim33-dependent transcriptome revealed targets of the transcription factor E2f4 as the most enriched group of deregulated genes (Fig. 2a). Since E2f factors are the key regulators of DNA replication, we pursued the idea that Trim33 regulates E2f4 abundance and activity. Immuno-fluorescence experiments showed an accumulation and preferential nuclear localization of E2f4 in Trim33-KO cells (Supplementary Fig. 2d, e). Analysis of protein turnover using cycloheximide chase assays showed that deletion of Trim33 stabilized E2f4, but not the related proteins E2f1, E2f3, or E2f5 (Fig. 2b; Supplementary Fig. 2f). Expression of wildtype but not of catalytically inactive Trim33 promoted E2f4 ubiquitination (Fig. 2c). Immunoprecipitation analysis and proximity ligation assays (PLA) showed interaction of endogenous Trim33 and E2f4, which was enhanced by inhibition of the proteasome (Fig. 2d, e). E2f4 was also readily detected in Trim33 immunoprecipitates from lysates of U2OS cells, in contrast to E2f1 and E2f3 (Supplementary Fig. 2g). In immunoprecipitation experiments with ectopic proteins, mutation of residues that mediate DNA binding (RRIYD65AAIAA−E2f4-ΔDB)[25], impaired the Trim33-E2f4 interaction (Fig. 2f, g). In contrast, mutation of lysine residues to arginine (E2f4-KR) or deletion of the C-terminal region (amino acids 331-414) that includes the pocket protein-binding domain (E2f4-ΔC), appeared to stimulate Trim33 recruitment. Consistently, GST-pulldown assays showed that the N-terminal fragment of E2f4 (amino acids 1–105) that includes the DNA binding domain, is sufficient for Trim33 recruitment (Fig. 2f, h). Mutation of conserved residues in the PHD-Bromo domain of Trim33 that mediate histone recognition (ED887AA, FN1038AA)[26,27], also impaired Trim33-E2f4 interaction (Supplementary Fig. 2h), indicating that Trim33 associates with E2f4 on chromatin. Supporting this idea, the treatment of whole cell lysates with benzonase facilitated the recovery of the Trim33-E2f4 complexes (Supplementary Fig. 2i).

siRNA-mediated depletion of E2f4 slowed down DNA replication forks during recovery from HU treatment in Trim33-KO cells (Fig. 2i; Supplementary Fig. 2j). Expression of E2f4 variants that lack DNA

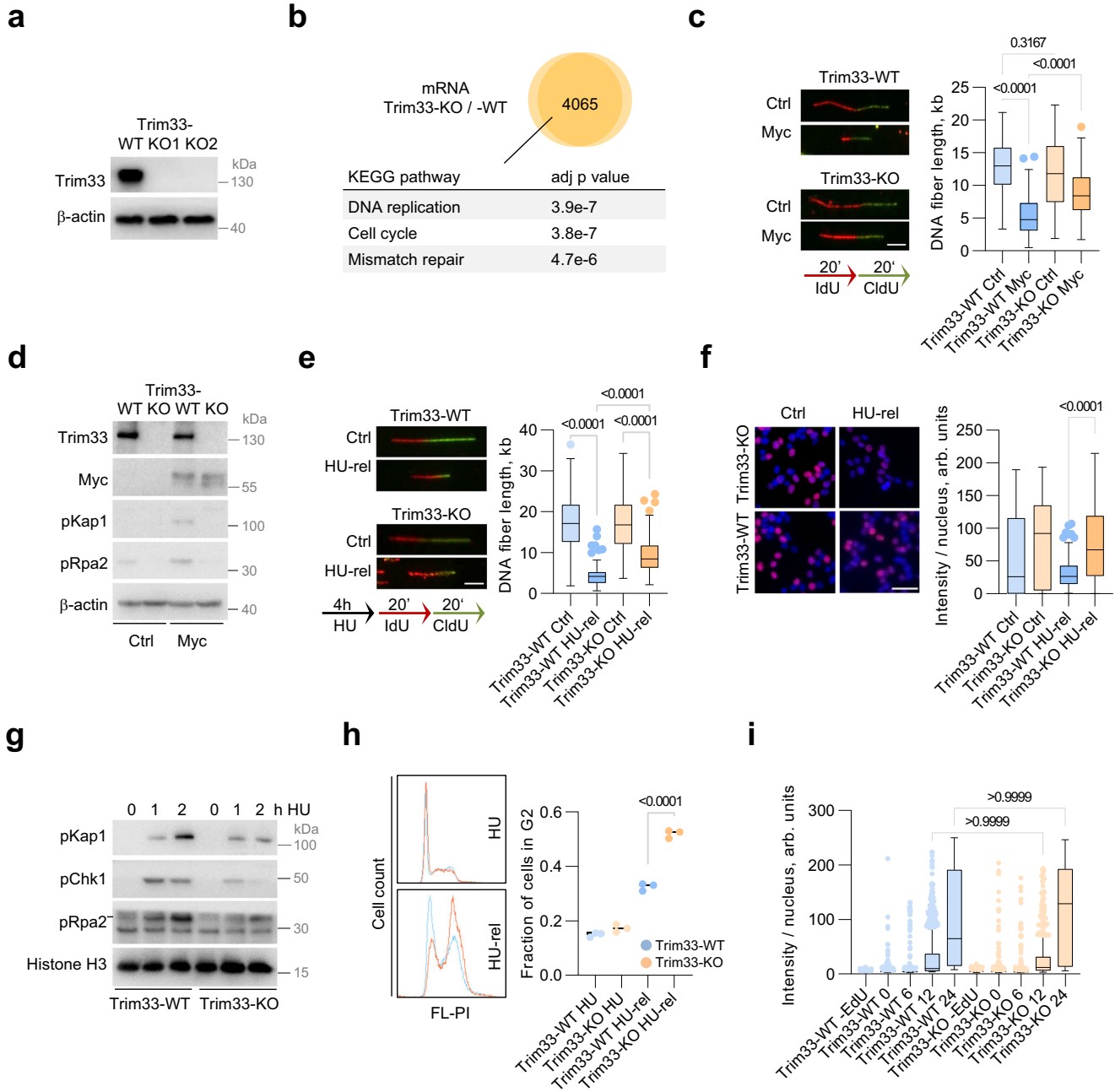

**Fig. 1 | Deletion of Trim33 abates replicative stress. a** Immunoblotting analysis of p19/Nras Trim33-WT and Trim33-KO cell lines; $n = 3$. **b** Analysis of the RNA-seq data for two Trim33-KO cell lines; $n = 3$. Venn diagram shows the number of commonly deregulated genes ($p$ adj <0.01, any log2FC) for the two knockout cell lines compared to Trim33-WT cells. Top three enriched KEGG pathways are shown (analyzed by Enrichr[70]). **c** DNA fiber assays in Trim33-WT and Trim33-KO cells expressing Myc or a Ctrl vector; $n = 2$. 100 fibers were measured per sample; scale bar = 1 μm. Significance was determined using Kruskal–Wallis test and Dunn's multiple comparisons. **d** Immunoblotting analysis in Trim33-WT and Trim33-KO cells expressing Myc or Ctrl vector; $n = 3$. **e** DNA fiber assays in Trim33-WT and Trim33-KO cells, unchallenged (Ctrl) or released from a 4-h hydroxyurea (HU) treatment (HU-rel); $n = 3$. 100 fibers were measured per sample; significance was determined as in **c**). Scale bar = 1 μm. **f** EdU incorporation analysis in Ctrl and HU-treated Trim33-WT

and Trim33-KO cells; $n = 2$. >240 cells per group were quantified; scale bar = 50 μm. Significance was determined as in **c**). **g** Immunoblotting analysis in Trim33-WT and Trim33-KO cells at indicated timepoints following release from HU; $n = 3$. **h** FACS analysis of PI-stained Trim33-WT and Trim33-KO cells treated with HU for 12 h, and after a 12 h release from the treatment. The graph shows an average of three technical replicates. Significance was determined using one-way ANOVA and Tukey's multiple comparisons. **i** Analysis of EdU incorporation during release from synchronization in G1 by serum deprivation; $n = 1$. At least 416 cells were analyzed per sample; the data were analyzed using Kruskal–Wallis test with Dunn's multiple comparisons. **c, e, f, i** Boxplots represent median±quartiles with whiskers ranging up to 1.5-fold of the interquartile range. Source data are provided as a Source Data file.

binding domain or the C-terminal region, also slowed replication forks and equalized fork progression in Trim33-WT and Trim33-KO cells after release from HU (Fig. 2j; Supplementary Fig. 2k), arguing that stabilization of E2f4 underlies accelerated DNA replication under stress upon loss of Trim33.

## Trim33 deletion promotes the association of E2f4 with the Recql helicase

We next sought to identify E2f4 targets with a function in controlling DNA replication under stress. Analysis of E2f4 genomic binding using ChIP-seq showed a strong increase in E2f4 recruitment to chromatin

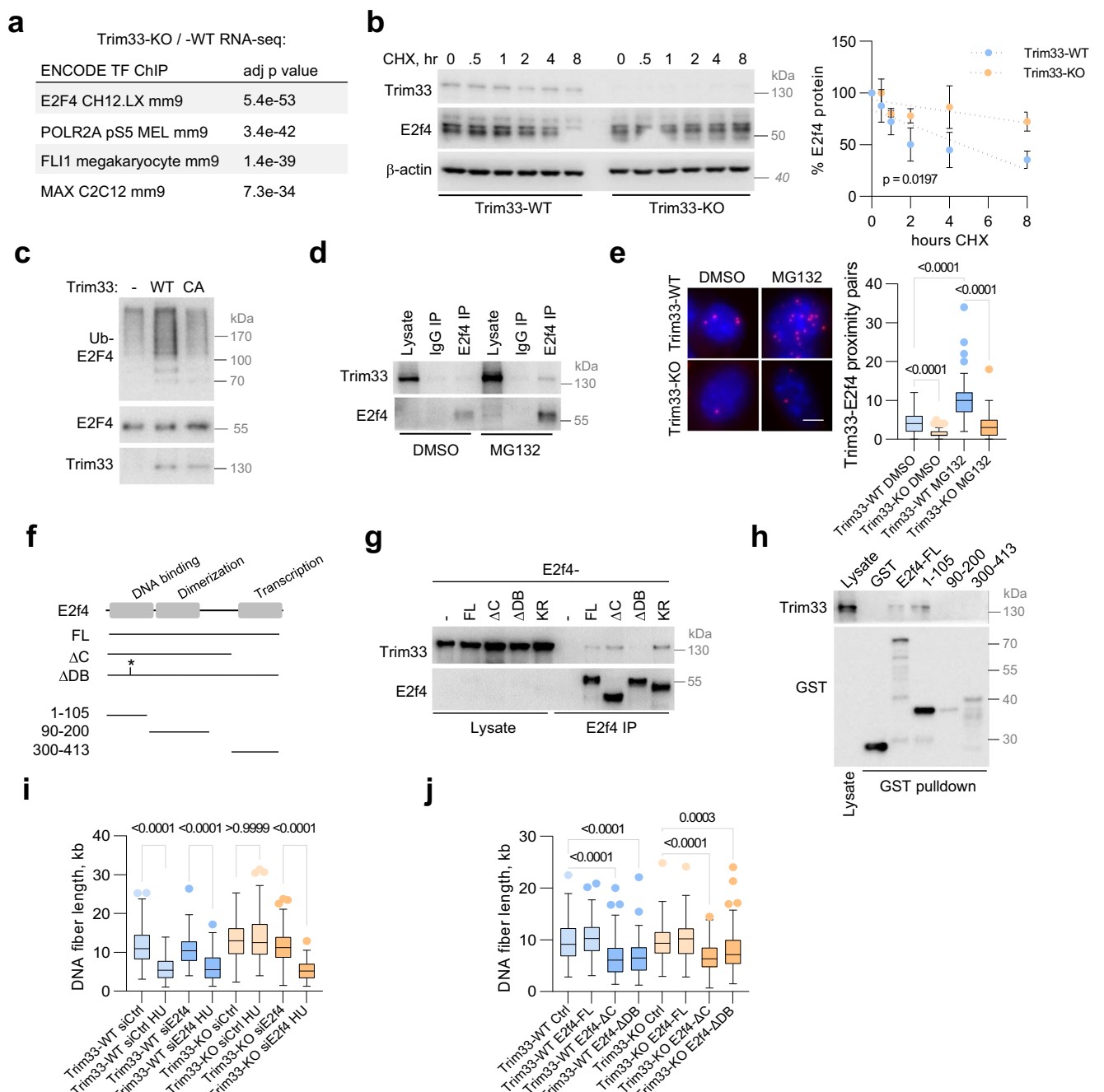

**Fig. 2 | Trim33 targets E2f4 to limit DNA replication under stress. a** The top significantly represented transcription factors targeting Trim33-deregulated genes in p19/Nras cells; data were analyzed using Enrichr[70]. **b** Cycloheximide chase assays in p19/Nras Trim33-WT and Trim33-KO cells; $n = 3$. Right panel shows the mean intensity (±SD) for three independent experiments with linear regression analysis. **c** Ubiquitin pulldown assays in HeLa cells transfected with vectors encoding E2f4, his-Ub, and wildtype (WT) or catalytically inactive (CA) Trim33; $n = 3$. **d** Immunoprecipitation assays with E2f4 antibodies in p19/Nras Trim33-WT cells treated with MG132 or DMSO; $n = 3$. **e** PLA assays in p19/Nras Trim33-WT with antibodies to Trim33 and E2f4. 50 cells per group were analyzed; $n = 3$. Scale bar = 2 μm. **f** Schematic of E2f4 domain structure and the variants used in the

following experiments. **g** Immunoprecipitation analysis following transient transfection of the indicated E2f4 variants in HeLa cells; $n = 2$. **h** GST-pulldown assays with the indicated GST-E2f4 fusion proteins and U2OS whole cell lysates; $n = 3$. **i** DNA fiber assays in p19/Nras Trim33-WT and Trim33-KO cells transfected with siRNA against E2f4 or a non-targeting control. 100 fibers were analyzed per condition; $n = 2$. **j** DNA fiber assays in p19/Nras Trim33-WT and Trim33-KO cells, expressing indicated E2f4 variants, after release from a 4 h HU treatment. 140 fibers were counted per condition; $n = 2$. **e, i, j** Significance was determined by Kruskal–Wallis test with Dunn's multiple comparisons. **e, i, j** Boxplots represent median±quartiles with whiskers ranging up to 1.5-fold of the interquartile range. Source data are provided as a Source Data file.

(mostly transcription start sites) in the absence of Trim33 (Fig. 3a, b). However, for most E2f4 binding sites, enhanced E2f4 recruitment was not associated with altered gene expression–less than 20% of E2f4-bound genes in Trim33-KO cells (363 from 2269) were significantly deregulated. For the repressed genes, there was a weak correlation

between E2f4 binding (number of ChIP-seq tags) and altered expression levels (log2FC) (Spearman correlation $r = -0.21$); for the upregulated genes there was no correlation (Spearman $r = -0.01$) (Fig. 3c). Most Trim33-dependent genes were regulated moderately (median log2FC of 0.58 and −0.56 for up- and downregulated genes) and

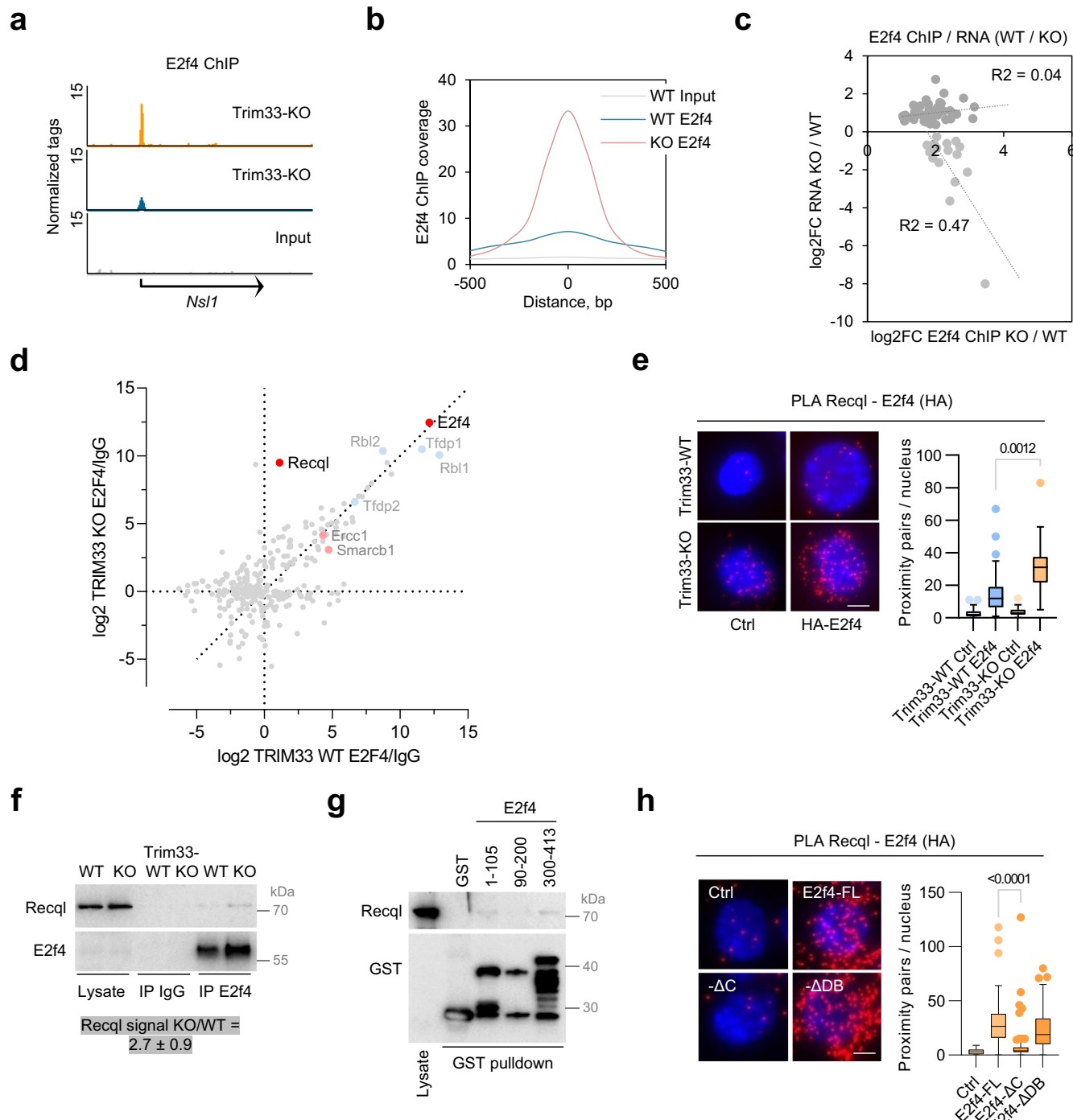

**Fig. 3 | Deletion of Trim33 promotes E2f4 interactions with chromatin and with the Recql helicase. a** Genome browser tracks of E2f4 ChIP-seq in p19/Nras Trim33-WT and Trim33-KO cells; $n = 2$. **b** Quantification of ChIP-seq tag coverage at E2f4 peaks in p19/Nras Trim33-WT and Trim33-KO cells. **c** Relationship between changes (log2FC) in E2f4 binding (normalized E2f4 ChIP-seq tags) and mRNA expression (log2FC) of the encoded gene for p19/Nras Trim33-KO vs Trim33-WT cells. Each dot is a bin of 5 genes sorted by the change in mRNA levels. Trendlines are derived from linear regression analysis. **d** LC-MS/MS analysis of E2f4 interactome in p19/Nras Trim33-WT and Trim33-KO cells. Known interaction partners are highlighted in blue. **e** PLA analysis of Recql-E2f4 association in p19/Nras Trim33-WT and Trim33-KO cells stably expressing HA-tagged E2f4 or vector Ctrl; from left, $n = 49,49,50,50$ cells; scale bar = 2 µm; $n = 3$ independent experiments. **f** Immunoprecipitation of endogenous E2f4 from benzonase-treated lysates of p19/Nras Trim33-WT and Trim33-KO cells using a mouse anti-E2f4 antibody. Quantification of three biological replicates shows mean intensity and SD. **g** GST-pulldown assay with the indicated GST-E2f4 fusion proteins and shTrim33 U2OS whole cell lysates; $n = 3$. **h** PLA analysis of Recql-E2f4 association in p19/Nras Trim33-KO cells stably expressing the indicated HA-tagged E2f4 variants; from left, $n = 100,100,97,100$ cells; scale bar = 2 µm; $n = 2$ experiments. **e, h** Significance was determined by Kruskal–Wallis test with Dunn's multiple comparisons. **e, h** Boxplots represent median±quartiles with whiskers ranging up to 1.5-fold of the interquartile range. Source data are provided as a Source Data file.

virtually all DNA replication genes were upregulated (Supplementary Fig. 3a). Furthermore, protein levels of several key replication proteins encoded by bound and upregulated genes, including subunits of the Mcm helicase and Cdc6, were similar in Trim33-WT and Trim33-KO cells (Supplementary Fig. 3b), suggesting that regulation at the protein level can diminish transcriptional effects. Notably, deletion of Trim33 and overexpression of Myc similarly deregulated the expression of genes, associated with DNA replication, but had opposite effects on replication fork progression (Supplementary Fig. 3c; Fig. 1c). Since E2f4 also has non-transcriptional functions[28], we considered the possibility that Trim33 and E2f4 regulate DNA replication independently of target gene expression.

To identify interaction partners of E2f4 in Trim33-WT and Trim33-KO cells we immunoprecipitated endogenous E2f4 from total cell lysates and analyzed these precipitates by LC-MS/MS. In addition to multiple known E2f4 interactors (e.g., Rbl1/2, Dp1/2, Lin9/54), this analysis identified several proteins, involved in response to replicative stress and DNA damage, including the Recql (Recq1) DNA helicase, proteins of the Swi-Snf family of chromatin remodeling complexes (Arid1a and Ini1) and the Ercc1-Xpf structure-specific nuclease (Fig. 3d). Recql was shown to stabilize stalled replication forks and mediate fork restart during release from replicative stress[29–31], leading us to analyze the functional link between E2f4 and Recql.

PLA and immunoprecipitation experiments confirmed that loss of Trim33 promoted Recql interaction with endogenous and stably expressed E2f4 (Fig. 3e, f). In contrast, the interaction of E2f4 with Ini1 and Ercc1 was not stimulated by the deletion of Trim33 (Supplementary Fig. 3d). GST-pulldown experiments with E2f4 fragments showed preferential E2f4 recruitment to the C-terminal transactivation domain of E2f4, although a very faint signal was also visible for the N-terminal region (Fig. 3g). In PLA and IP experiments deletion of the C-terminal region of E2f4 reduced or abolished Recql recruitment (Fig. 3h; Supplementary Fig. 3e), indicating that in cells the E2f4-Recql interaction is primarily mediated by the transactivation domain of E2f4.

## E2f4 recruits Recql to promote DNA replication in Trim33-KO cells

To assess the association of Recql with chromatin in Trim33-WT and Trim33-KO cells, we first performed PLA assays with antibodies to endogenous Recql and either RNAPII or Mcm2, a subunit of the replicative helicase. Deletion of Trim33 strongly enhanced association of Recql with RNAPII, indicating recruitment to transcriptionally active loci, and with Mcm2, indicating recruitment to the replisome (Supplementary Fig. 4a). This idea was supported using immunoprecipitation of HA-tagged Recql from formaldehyde-crosslinked cells, which showed an increased association of Recql with RNAPII and Mcm2 in Trim33-KO cells (Fig. 4a).

To compare the association of Recql with chromatin in Trim33-WT and Trim33-KO cells at the genomic scale, we performed Cut&Run assays[32] in formaldehyde-fixed p19/Nras cells, stably expressing low levels of HA-tagged Recql. Chromatin binding of Recql was strongly stimulated by deletion of Trim33 (Fig. 4b, c; Supplementary Fig. 4b), with a more than 10-fold increase in the number of binding sites (9413 in Trim33-KO cells vs 616 in Trim33-WT cells). Recql binding was particularly enriched at E2f4-bound sites (Fig. 4c), suggesting that E2f4 promotes Recql recruitment to chromatin.

To test this idea, we immunoprecipitated endogenous Recql from crosslinked Trim33-WT and Trim33-KO cells, transfected with Control or E2f4 siRNA. Depletion of E2f4 strongly reduced the association of Recql with RNAPII, Mcm2, and PCNA (Fig. 4d). Consistently, Cut&Run-PCR assays with Recql antibodies showed a strongly diminished association of Recql with several tested promoter regions in Trim33-KO cells expressing E2f4 siRNA (Fig. 4e).

Expression of E2f4 variants that are deficient in Recql or DNA binding (E2f4-ΔC and E2f4-ΔDB), also diminished the association of

endogenous Recql with RNAPII, Mcm2, and PCNA in Trim33-KO cells (Supplementary Fig. 4c, d). Analysis of genomic binding of endogenous Recql in these cells using Cut&Run assays with Recql antibodies showed a global reduction in Recql recruitment to chromatin upon expression of E2f4-ΔC or E2f4-ΔDB (Fig. 4f, g).

Depletion of Recql using siRNA slowed replication fork progression after release from HU in Trim33-KO cells (Fig. 4h, i). Similarly, expression of a helicase-deficient variant Recql-K119A[33] also slowed replication forks in Trim33-KO cells (Supplementary Fig. 4e, f), arguing that ectopic recruitment of Recql underlies accelerated DNA replication in the absence of Trim33.

## Recql and E2f4 are recruited to chromatin upon replicative stress

In line with previous observations[30,32], immunoprecipitation assays showed robust recruitment of endogenous Recql to replisome proteins (Mcm2 and PCNA), but also to RNAPII, upon HU-induced fork stalling (Fig. 5a). To test whether Recql is associated with replication forks that restart upon release from stress, we labeled nascent DNA with thymidine analog EdU for 10 min after release from 4 h HU treatment[33]. EdU-labeled DNA was biotinylated in a click reaction with biotin-azide and PLA assays were performed with antibodies to biotin and Recql. This experiment showed an association of Recql with nascent DNA in Trim33-WT cells, which was strongly increased in Trim33-KO cells (Fig. 5b). Capture of biotin-dUTP-labeled nascent chromatin after release from HU[34] revealed an increased association of stably expressed HA-tagged Recql, as well as of RNAPII and PCNA with nascent DNA in Trim33-KO cells, compared to control cells (Supplementary Fig. 5a), consistent with the idea that Recql facilitates DNA synthesis at transcriptionally active loci during release from replication stress.

Cut&Run assays showed that HU induced robust recruitment of HA-tagged Recql to chromatin in Trim33-WT cells, which was especially prominent at E2f4-bound sites (Fig. 5c, d; Supplementary Fig. 5b, c). This effect was abolished in Trim33-KO cells, suggesting that Trim33-mediated regulation of E2f4 is compromised under replicative stress to promote Recql binding to chromatin. In line with this idea, HU diminished ubiquitination of E2f4 by Trim33 (Fig. 5e) and impaired Trim33-E2f4 interaction, as shown by immunoprecipitation and PLA assays (Fig. 5f, g). Consistently, HU and other genotoxins that induce fork stalling and activate checkpoint signaling, including etoposide and doxorubicin, upregulated E2f4 protein levels and induced Recql recruitment to RNAPII (Supplementary Fig. 5d, e). Induction of Myc in U2OS cells also upregulated E2f4 levels and promoted E2f4 association with RNAPII and with Recql (Supplementary Fig. 5f), suggesting that stabilization of E2f4 and recruitment of Recql are a general response to checkpoint activation.

Cut&Run experiments revealed a strong increase in genomic binding of endogenous E2f4 upon HU treatment in p19/Nras cells with 1295 peaks in HU-treated cells compared to 534 in unchallenged cells (Fig. 5h). Transcriptional profiling of HU-treated and control cells using RNA-seq showed that E2f4 recruitment was largely uncoupled from changes in gene expression−gene set enrichment analysis revealed that HU primarily deregulated transcriptional targets of CEBP, REST and p53 transcription factors (Supplementary Fig. 5g). Of 2000 genomic positions, which showed enhanced E2f4 binding upon HU, only 183 were in significantly deregulated genes and for those, E2f4 binding did not correlate with mRNA levels (Supplementary Fig. 5h). In contrast, HU-induced E2f4 binding showed a good correlation with Recql recruitment (Spearman $r = 0,5996$) (Fig. 5i). In accord, depletion of E2f4 using siRNA diminished HU-induced Recql recruitment at several tested transcriptional start sites (Supplementary Fig. 5i).

E2f4 variants, deficient in Recql recruitment or DNA binding, and a helicase-dead Recql variant K119A slowed the progression of replication forks in Trim33-WT cells after release from HU (Supplementary

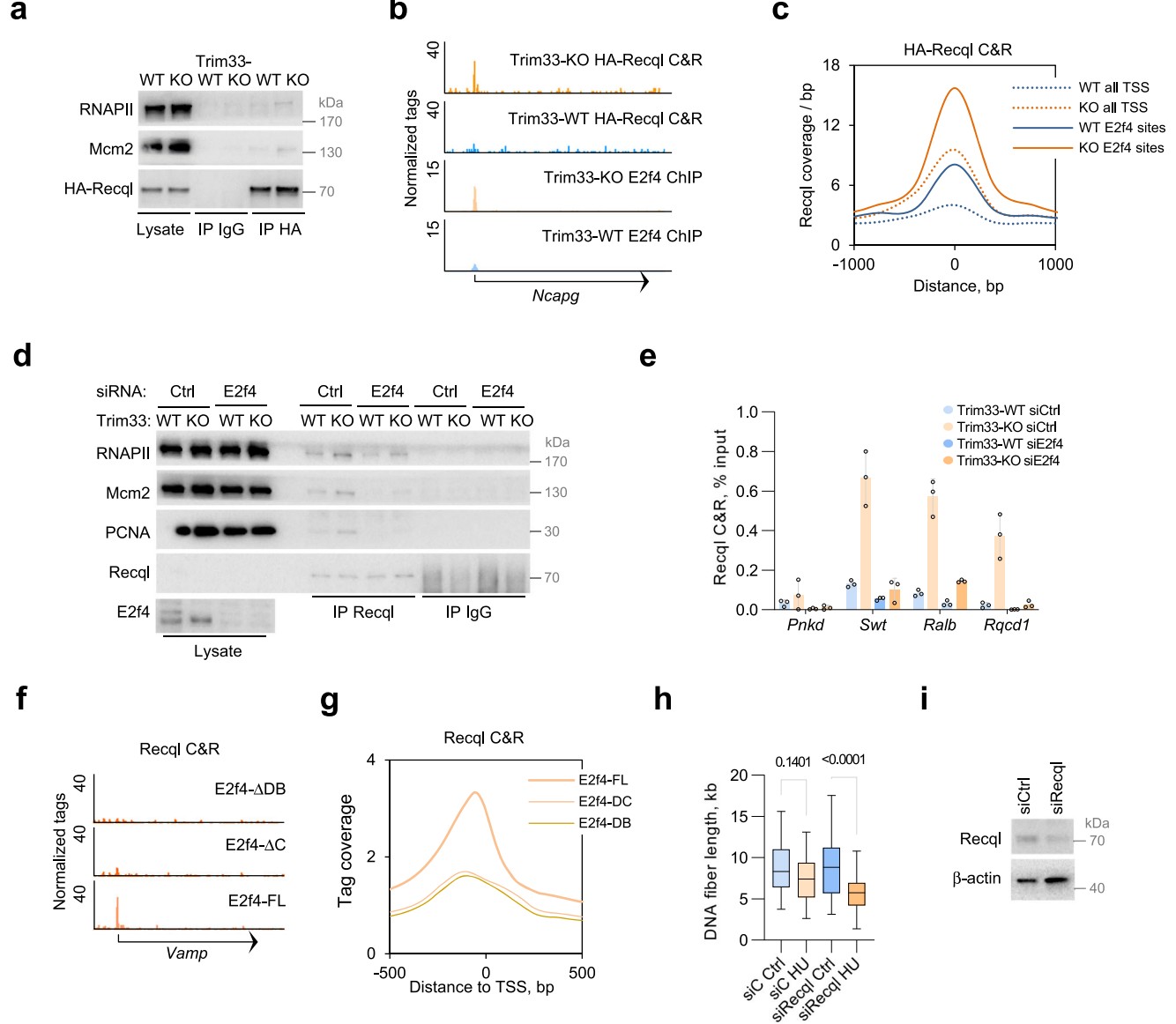

**Fig. 4 | E2f4 promotes Recql binding to chromatin in Trim33-KO cells.**
**a** Immunoprecipitation from p19/Nras Trim33-WT and Trim33-KO cells expressing HA-tagged Recql; *n* = 3. **b** Representative genome browser tracks of Cut&Run analysis of HA-Recql binding in p19/Nras Trim33-WT and Trim33-KO cells expressing HA-tagged Recql. Lower two tracks−E2f4 ChIP-seq. **c** Sequencing tag coverage for Recql Cut&Run signal from the experiment shown in **b**) at all TSS and at E2f4 sites. **d** Immunoprecipitation analysis with Recql antibodies from formaldehyde-crosslinked p19/Nras Trim33-WT and Trim33-KO cells, transfected with E2f4-targeting or a control siRNA. **e** Cut&Run assays with Recql antibodies in p19/Nras Trim33-WT and Trim33-KO cells followed by qPCR analysis at the TSS sites of indicated genes. Mean values of percent input for three technical replicates and standard deviation are plotted. **f** Representative genome browser tracks of the Cut&Run assays with Recql antibodies in Trim33-KO p19/Nras cells expressing the indicated E2f4 variants. **g** Cut&Run sequencing tag coverage at all TSS for the experiment shown in **f**). **h** DNA fiber assays in Trim33-KO cells, expressing siRNA against Recql or control siRNA, untreated or released from a 4 h HU treatment; *n* = 2. 100 fibers were counted per sample and the data were analyzed using Kruskal−Wallis test followed by Dunn's multiple comparison. Boxplots represent median±quartiles with whiskers ranging up to 1.5-fold of the interquartile range. **i** Immunoblotting analysis of Recql expression in Trim33-KO cells, used in **h**). Source data are provided as a Source Data file.

Fig. 5j, k, l), arguing that recruitment of Recql by E2f4 facilitates DNA synthesis during recovery from stress. Recent studies suggest that Recql can promote the restart of replication forks after collisions with RNAPII (transcription-replication conflicts, TRCs)[35], predicting a lower incidence of TRCs in Trim33-KO cells. In support of this idea, PLA experiments with antibodies to RNAPII and PCNA[36] revealed strongly reduced levels of TRCs in Trim33-KO cells for both initiating (S5-phosphorylated) and total RNAPII (Fig. 5j). Collisions for pS5-RNAPII were reduced upon HU treatment, in line with pronounced recruitment of Recql to chromatin under these conditions. In contrast, collisions for total RNAPII did not decrease upon HU in Trim33-WT cells and inversely correlated with a much broader distribution of Recql on chromatin and accelerated recovery of DNA replication in Trim33-deficient cells.

**Trim33 limits replication-associated DNA damage under stress**
As deregulated DNA replication is a prime source of genomic instability, we assessed DNA damage in cells cultured in the presence of low concentrations of hydroxyurea. Deletion of Trim33 strongly elevated levels pf pH2AX, a marker of DNA double-strand breaks

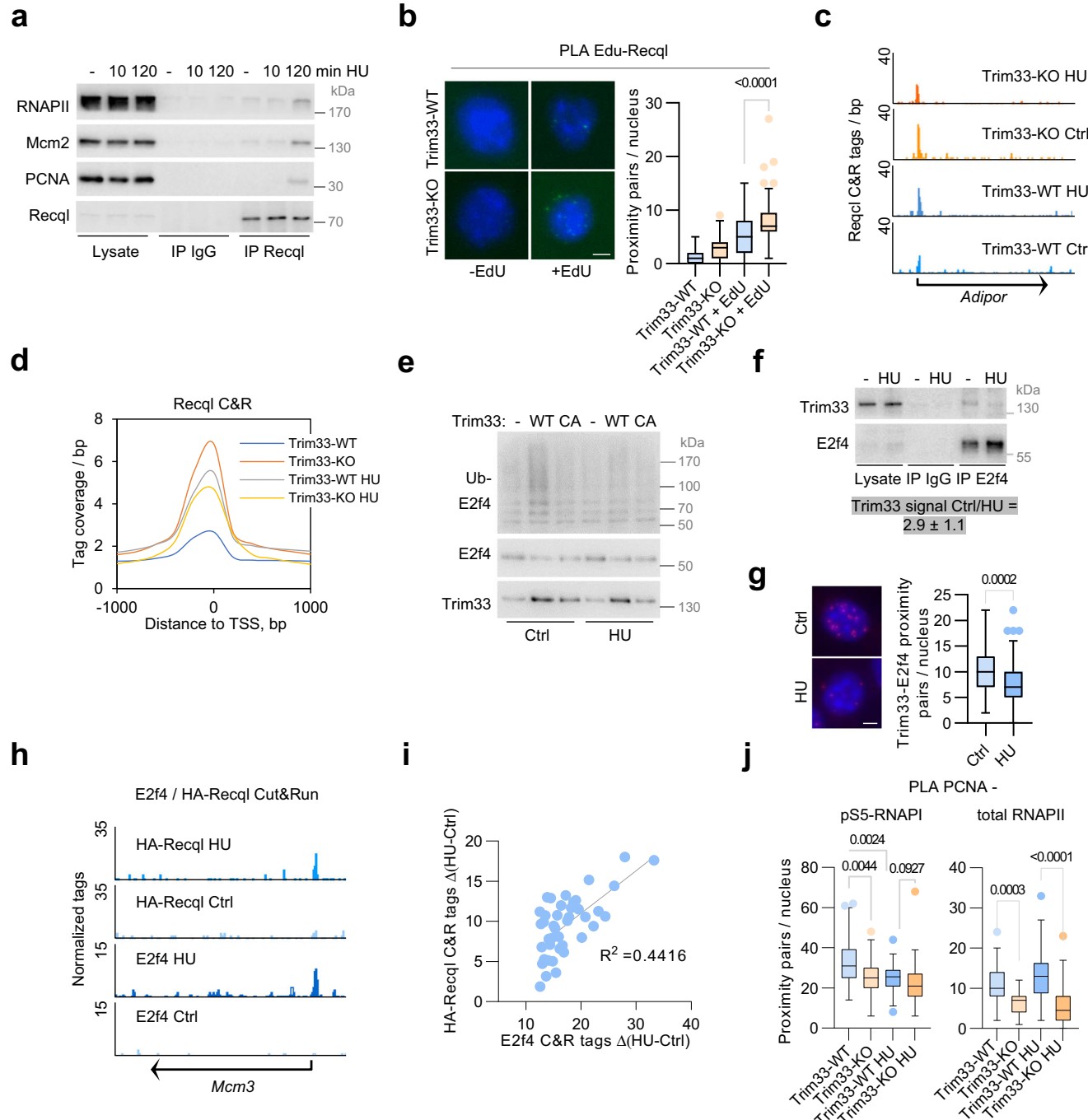

**Fig. 5 | Replicative stress triggers Recql recruitment to chromatin.**
**a** Immunoprecipitation of endogenous Recql from formaldehyde-crosslinked p19/
Nras Trim33-WT cells, untreated or treated with HU for 10 or 120 min; $n = 3$. **b** PLA
assays with antibodies to biotinylated EdU-labeled nascent DNA and Recql in
Trim33-WT and Trim33-KO cells after release from a 4 h HU treatment; from left,
$n = 112,62,143,113$ cells; scale bar = 5 μm; $n = 3$ experiments. Significance was deter-
mined using Kruskal–Wallis test with Dunn's multiple comparisons.
**c** Representative genome browser tracks of HA-Recql Cut&Run in p19/Nras Trim33-
WT and Trim33-KO cells, untreated or treated with HU for 4 h. **d** Sequencing tag
coverage for Recql Cut&Run experiment shown in **c**). **e** Ubiquitin pulldown assay in
HeLa cells, transfected with vectors for His-Ub, HA-tagged E2f4, and Trim33-WT or
Trim33-CA; $n = 2$. **f** Immunoprecipitation of endogenous E2f4 from benzonase-
treated lysates of control or HU-treated (2 h) p19/Nras Trim33-WT cells. Quantifi-
cation of four independent experiments is shown as mean and S.D. of signal

intensity **g** PLA assays with Trim33 and E2f4 antibodies in untreated and HU-treated
(2 h) p19/Nras Trim33-WT cells. Significance was determined by a two-tailed *t*-test.
From left, $n = 98,117$ cells; Scale bar = 5 μm. **h** Representative genome browser
tracks of Cut&Run assay with E2f4 antibodies in untreated and HU-treated (4 h)
p19/Nras Trim33-WT cells, expressing HA-Recql. The upper two tracks show HA-
Recql binding in the same cells. **i** Relationship between E2f4 and Recql recruitment
at 2000 transcription start sites with maximal E2f4 enrichment after HU treatment
in Trim33-WT cells. The regions were sorted by E2f4 tags and binned at 50 genes/
bin followed by linear regression analysis. **j** PLA assay with antibodies to PCNA and
pS5-RNAPII or total RNAPII in untreated or HU-treated (4 h) p19/Nras Trim33-WT
and Trim33-KO cells. From left, $n = 50,50,50,46,39,46,50,50$ cells. Significance was
determined using Kruskal–Wallis test with Dunn's multiple comparisons.
**b**, **g**, **j** Boxplots represent median±quartiles with whiskers ranging up to 1.5-fold of
the interquartile range. Source data are provided as a Source Data file.

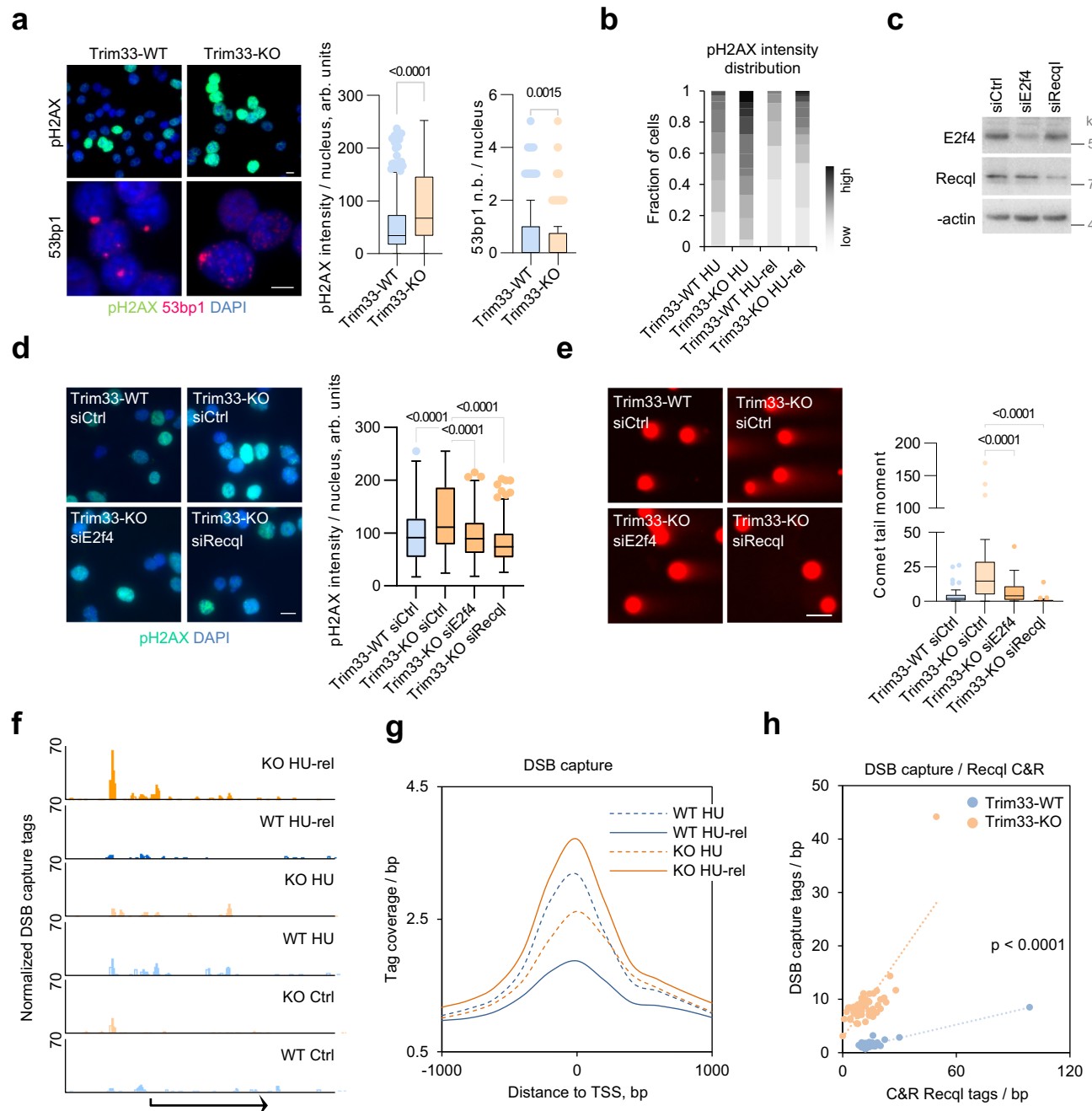

**Fig. 6 | Trim33 loss exacerbates DNA damage following release from replicative stress. a** Immunofluorescence analysis with pH2AX and 53bp1 antibodies in p19/Nras Trim33-WT and Trim33-KO cells cultured in the presence of 0.1 mM HU. Scale bar = 5 μm. Graphs show pH2AX signal intensity and 53bp1 nuclear bodies (n.b.) per nucleus for *n* = 231,228 cells (Trim33-WT,KO). Significance was determined by a two-tailed Mann–Whitney test. A representative of two independent experiments is shown. **b** Distribution of pH2AX signal intensity in p19/Nras Trim33-WT and Trim33-KO cells, upon 24 h exposure to HU (HU) or after a 24-h release from HU treatment (HU-rel). **c** Immunoblots of p19/Nras Trim33-KO cells transfected with the indicated siRNAs; *n* = 3. **d** Immunofluorescence staining with pH2AX antibody in p19/Nras Trim33-WT and Trim33-KO cells expressing the indicated siRNAs 24 h after release from HU. From left, *n* = 161,219,97,139 cells; significance was determined using Kruskal–Wallis test with Dunn's multiple comparisons; scale bar = 20 μm. **e** Neutral

comet assays in p19/Nras Trim33-WT and Trim33-KO cells transfected with the indicated siRNAs, untreated or released from HU treatment as in **d**). Scale bar = 20 μm. From left, *n* = 59,34,64,60 cells. Significance was determined by one-way ANOVA with Tukey's multiple comparisons. **f** Representative genome browser tracks of DSB capture and sequencing signal in untreated (Ctrl), HU-treated (HU), or HU-treated and released (HU-rel) Trim33-WT and Trim33-KO cells. **g** DSB capture tag density at all TSS for HU-treated (24 h) or HU-treated and released for 24 h p19/Nras Trim33-WT and Trim33-KO cells. **h** Linear regression analysis of Recql recruitment (normalized HA-Recql Cut&Run tags for HU-treated cells) vs. DSB capture tags at 24 h after HU release for 3000 sites with strongest Recql binding, sorted by the number of Recql tags and binned at 60 sites/bin. **a**, **d**, **e** Boxplots represent median ± quartiles with whiskers ranging up to 1.5-fold of the inter-quartile range. Source data are provided as a Source Data file.

(DSBs) (Fig. 6a), in contrast to normal growth conditions (Supplementary Fig. 6a). Notably, the number of 53bp1 nuclear bodies, which incorporate under-replicated genomic loci[37–39], was lower in Trim33-KO cells (Fig. 6a; Supplementary Fig. 6b), indicating a defect in

recognition of such loci. Following release from replication arrest, induced by standard concentrations of HU (1 mM), Trim33-KO cells also sustained much higher levels of pH2AX compared to controls (Fig. 6b), pointing to an accumulation of DNA damage following

replicative stress. siRNA-mediated knockdown of E2f4 or Recql decreased pH2AX levels after release from HU in Trim33-KO cells (Fig. 6c, d). Similarly, expression of E2f4-ΔC, E2f-ΔDB, or Recql-K119A reduced pH2AX levels (Supplementary Fig. 6c, d) in Trim33-KO cells under these conditions.

Neutral comet assay revealed increased levels of DNA breakage after HU release in Trim33-KO cells relative to Trim33-WT cells, whereas there was no difference under normal growth conditions (Supplementary Fig. 6e). Knockdown of E2f4 or Recql suppressed DNA breakage in Trim33-KO cells (Fig. 6e), suggesting that ectopic recruitment of E2f4 and Recql underlies accumulation of DNA damage during recovery from stress in the absence of Trim33.

We then analyzed the effect of Trim33 deletion on the genome-wide distribution of DNA DSBs using DSB capture and sequencing[40]. In Trim33-WT cells, the DSB signal diminished following release from HU, indicative of DNA repair (Fig. 6f, g). In contrast, in Trim33-KO cells, the DSB signal increased after release from HU, suggesting an accumulation of DNA replication-associated DNA lesions. Notably, many HU-induced DSBs localized in the vicinity of E2f4- and Recql-bound transcription start sites (Supplementary Fig. 6f, g). DSB capture signal increased much stronger with Recql binding in HU-treated Trim33-KO cells (linear regression slope = 0.51), compared to Trim33-WT cells (slope = 0.08) (Fig. 6h), suggesting that in the absence of Trim33, rapid restart and progression of replication forks at E2f4 and Recql-bound sites during recovery from HU leads to DNA breakage. Consistent with the effects on DNA damage, knockout of Trim33 compromised long-term survival following release from HU (Supplementary Fig. 6h). Stable expression of E2f4-ΔDB or Recql-K119A promoted survival of Trim33-KO cells after HU treatment (Supplementary Fig. 6i), in line with reduced levels of DNA damage in these cells (Supplementary Fig. 6c, d).

### Loss of Trim33 exacerbates Myc-induced DNA damage and delays liver tumorigenesis

Since the depletion of Trim33 alleviated Myc-induced replicative stress (Fig. 1; Supplementary Fig. 1), we analyzed the impact of the Trim33-E2f4-Recql pathway on DNA damage in cells with high Myc levels. Stable expression of E2f4-FL but not of E2f4-ΔC, that does not bind Recql, accelerated replication fork progression in Myc-overexpressing cells (Supplementary Fig. 7a, b). Conversely, full-length E2f4-FL and Recql, but not E2f4-ΔC, E2f4-ΔDB, or Recql-K119A, increased pH2AX levels in these cells (Supplementary Fig. 7c, d), indicating that ectopic E2f4 and Recql activity can induce DNA damage upon Myc activation.

Depletion of Trim33 significantly increased Myc-induced pH2AX levels (Supplementary Fig. 7e), mimicking the effect of E2f4 and Recql overexpression. In contrast, Myc-induced increase in the number of 53bp1 bodies was reversed by depletion of Trim33 (Fig. 7a), in line with a reduced checkpoint signaling in Trim33-KO cells (Fig. 1d, g) and the data obtained with HU treatment (Fig. 6a). Neutral comet assay showed a small increase in DNA breakage upon Myc overexpression, which was strongly elevated by depletion of Trim33 (Supplementary Fig. 7f). The higher levels of DNA damage were accompanied by impaired colony formation upon knockdown of Trim33 in Myc-overexpressing cells (Supplementary Fig. 7g), similar to the phenotype found in HU-treated cells (Supplementary Fig. 6h).

siRNA-mediated depletion of E2f4 or Recql reduced pH2AX levels in Myc-overexpressing shTrim33 cells (Fig. 7b). Furthermore, comet assays showed that depletion of E2f4 or Recql reverted the increase in DNA breakage in Myc-overexpressing shTrim33 cells (Fig. 7c), indicating in the absence of Trim33, Myc-induced replicative stress evokes DNA damage via E2f4 and Recql.

To determine the impact of Trim33 on Myc-induced tumorigenesis, we used an autochthonous model of liver cancer[41] (Fig. 7d). Transposons, expressing Ras and Myc with shRNA targeting Trim33 or a control sequence (shRen)[23], were delivered using hydrodynamic tail

vein injection and mice were monitored for tumor development. Co-expression of Myc and shTrim33 delayed tumor development and significantly extended animal survival (Fig. 7e, f). We then assessed the expression of Trim33 and pH2AX using immunohistochemistry on tissue sections of tumor-bearing livers. Trim33-deficient tumors had strongly elevated levels of pH2AX (Fig. 7g, h), suggesting that DNA damage upon acute Myc overexpression delays tumor development in the liver. Since replicative stress in tumors is associated with the recruitment of immune cells[42], we analyzed the tumor samples by immunohistochemistry with antibody to CD3, which marks T lymphocytes. Trim33-deficient tumors had a significantly increased number of CD3-positive cells (Supplementary Fig. 7h), suggesting that loss of Trim33 can promote tumor infiltration by T cells, contributing to delayed tumor development. We concluded that via regulation of E2f4 and Recql, Trim33 limits DNA damage in Myc-overexpressing cells and thereby promotes Myc-driven tumorigenesis.

## Discussion

In this study, we show that the Trim33 ubiquitin ligase restricts interactions of the transcription factor E2f4 with the DNA helicase Recql to control the progression of DNA replication forks under replicative stress. Deletion of Trim33 triggers ectopic recruitment of Recql to chromatin, leading to rapid DNA replication and delayed checkpoint signaling. Consequently, Trim33 deficiency leads to the accumulation of DNA damage, impairs survival under replicative stress, and delays Myc-dependent liver tumorigenesis.

Multiple studies have implicated Trim33 in the regulation of RNAPII-driven transcription. For example, Trim33 controls several transcription factors, including Smad2/3, β-catenin, and PU.1 via proteolytic or non-proteolytic mechanisms[43–46]. Furthermore, Trim33 can broadly affect RNAPII function via recruitment of elongation factors pTEFb and FACT to transcription start sites[47], in line with its genome-wide localization to chromatin[46]. Via the PHD-Bromo domain, Trim33 recognizes methylated and acetylated histones[27,45] with a distinct specificity compared to two homologous proteins—Trim24 and Trim28, which form complexes with Trim33[48]. Intriguingly, chromatin recruitment activates the ubiquitin ligase function of Trim33, providing a mechanism for selective regulation of DNA-bound factors[27].

Extending these studies, we show that Trim33 associates with the transcription factor E2f4 and targets it for degradation. E2f4 is a member of the E2f family of transcription factors that play a key role in the regulation of cell cycle progression and DNA replication[49]. Trim33-E2f4 interaction requires the N-terminal region of E2f4, which includes the DNA-binding domain. Accordingly, mutations in either E2f4 or Trim33 that prevent DNA binding, impair their interaction, suggesting that Trim33 regulates a chromatin-associated pool of E2f4. Interestingly, previous studies revealed an interaction between E2f4 and Trim28[50], indicating that E2f4 may be regulated by oligomeric complexes containing Trim33.

Trim33 has also been implicated in the regulation of cellular response to DNA damage. Trim33 is recruited to DNA damage sites in a PARP1-dependent manner and promotes degradation of the chromatin remodeler Alc1[20]. Furthermore, depletion of Trim33 in several cell culture models sensitizes cells to ATR and Parp1 inhibitors[20,51]. E2f4, as a novel substrate of Trim33, can potentially integrate the different functions of Trim33 in transcription and the DNA damage response. Originally E2f4 was identified as a transcriptional regulator that mediates repression by pocket proteins and Smad factors[52,53]. Other studies showed that E2f4 can act as an activator that controls cell proliferation, DNA repair, and tumorigenesis[28,54]. E2f4 broadly binds chromatin, shares many binding sites, and can functionally cooperate with Myc and Nrf2[55,56]. Furthermore, E2f4 has non-transcriptional functions in cilial biogenesis and recruitment of Gemc1 and Multicilin to chromatin[57,58]. In line with these observations, we identify a number of factors, implicated in DNA replication and DNA repair, including

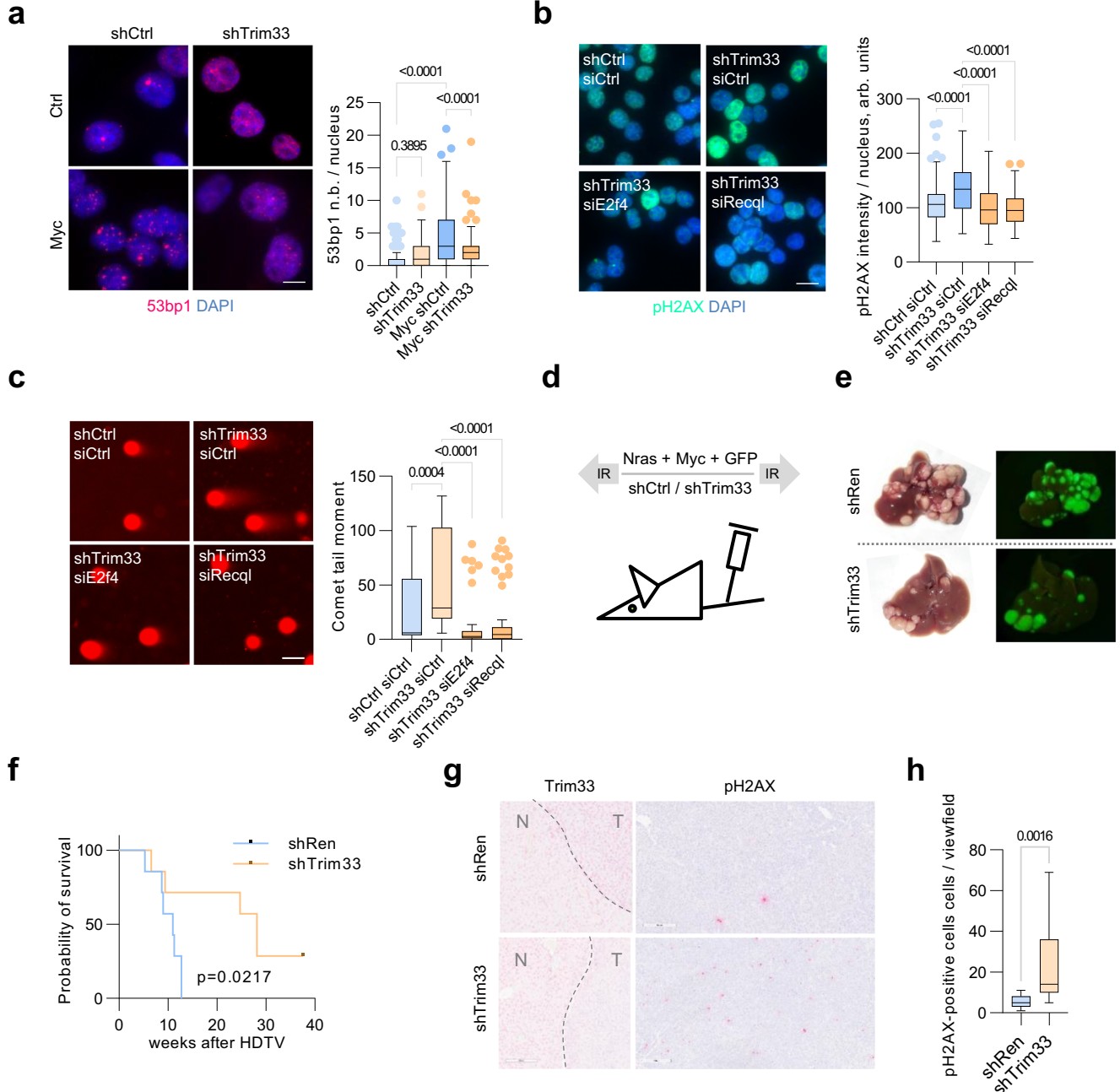

**Fig. 7 | Deletion of Trim33 promotes Myc-induced DNA damage and delays liver tumorigenesis. a** Immunofluorescence analysis with 53bp1 antibodies in p19/Nras (Ctrl) and p19/Nras/Myc (Myc) cells expressing shCtrl or shTrim33. At least 24 cells were quantified. Scale bar = 2 μm. **b** Immunofluorescence analysis with pH2AX antibodies in p19/Nras/Myc cells expressing shTrim33 or shCtrl and siRNAs against E2f4, Recql or control. At least 133 cells were quantified per group. Scale bar = 5 μm. **c** Neutral comet assays in p19/Nras/Myc cells expressing shRNAs against Trim33 or a control sequence and siRNAs against E2f4, Recql, or control. Scale bar = 10 μm. At least 39 cells were quantified per group. **d** Schematic of the HDTV-based model of liver tumorigenesis. Transposon vectors encoding Myc, Nras, GFP, and shCtrl or shTrim33 are injected along with the SB transposase in the tail vein of C57BL/J mice.

**e** Representative images of whole livers under daylight and ultraviolet light showing tumor nodules. **f** Kaplan–Meier survival plots for two cohorts of mice injected with shTrim33 or shCtrl (7 animals/group). Significance was determined using the log-rank test. **g** Representative images of IHC analysis with antibodies to Trim33 and pH2AX on FFPE sections of tumor-bearing livers. **h** Quantification of pH2AX-positive cells per view field in liver tumors expressing shTrim33 or shCtrl. Significance was determined by a two-tailed, unpaired *t*-test. **a**, **b**, **c** Significance was determined using Kruskal–Wallis test with Dunn's multiple comparisons.
**a**, **b**, **c**, **h** Boxplots represent median±quartiles with whiskers ranging up to 1.5-fold of the interquartile range. Source data are provided as a Source Data file.

components of the Swi/Snf and Ercc1/4 complexes and the Recql DNA helicase, as E2f4 interacting partners.

Recql is the most abundant Recq family helicase in mammalian cells, implicated in the cellular response to replicative stress[29,30,59]. In contrast to other Recq helicases, such as Blm and Wrn, Recql does not resolve R-loops but mediates DNA strand separation and promotes

branch migration, which likely underlies its function in promoting the restart of stalled replication forks[60,61]. This function of Recql can antagonize Parp1 activity in the stabilization of reversed replication forks[29] and possibly accounts for the rapid DNA replication observed in Parp1-inhibited cells[5] and in Trim33-KO cells, as described in this study. Recql, unlike some other Recq helicases, lacks the Helicase and RNase

D C-terminal DNA binding domain (HRDC)[62], suggesting that other proteins can facilitate Recql localization to stalled forks. For example, a recent report showed that the Foxa1 transcription factor can recruit Recql to the *ESR1* gene to stimulate its expression[63].

We show that E2f4 recruits Recql via the C-terminal domain, which mediates interaction with pocket proteins and the transcriptional function of E2f4. As pocket proteins interact with replicative helicase and modulate DNA replication[64], one interesting possibility is that E2f4 can be switched, upon activation of the replication checkpoint, from replication-inhibitory complexes with pocket proteins to permissive complexes that contain Recql.

Trim33 antagonizes the E2f4-Recql interaction and deletion of Trim33 leads to E2f4-dependent genome-wide localization of Recql to chromatin. Furthermore, the ability of E2f4 to recruit Recql is stimulated in response to hydroxyurea or other genotoxins that activate replication checkpoint signaling, demonstrating that it presents a general mechanism in the cellular response to replicative stress. The molecular underpinnings of E2f4-Trim33 complex regulation remain to be investigated. Trim33 may be modified by checkpoint kinases or by Parp1 to diminish recognition of E2f4 or prevent Trim33 association with histones to blunt ubiquitin ligase activity[27]; E2f4 may be also modified at the checkpoint to impair Trim33 binding. While replicative stress compromises Trim33-dependent regulation of E2f4 transiently, deletion Trim33 leads to pervasive Recql recruitment to chromatin. This aberrant localization of Recql likely underlies ectopic fork acceleration under stress and impairs the recognition and repair of replication-associated DNA lesions, as indicated by abated checkpoint signaling, inefficient formation of 53bp1 foci, and compromised DNA repair.

Upon stress, Recql is recruited to transcription start and termination sites, which are associated with RNAPII stalling, formation of R-loops, and transcription-replication conflicts[65–67]. This observation suggests that Recql can facilitate the progression of forks that stall during collisions with RNAPII or stimulate the firing of origins in the vicinity of transcription start sites, in line with previous observations[31,35,66]. Our data suggest that transient Recql recruitment in response to replicative stress primarily allows the bypass of conflicts that engage pS5-RNAPII, most likely at a pool of promoter-proximal sites. In contrast, complete loss of Trim33 results in a much broader distribution of Recql on chromatin and premature resolution of TRCs across the genome.

In line with the known functions of TRCs in the induction of ATM signaling[36], rapid resolution of TRCs and accelerated replication fork progression in Trim33-KO cells is paralleled by a blunted checkpoint activation upon replicative stress. This correlates with a diminished formation of 53bp1 nuclear bodies, which incorporate under-replicated genomic loci and require ATM signaling for assembly[37–39]. Since 53bp1 controls DNA replication and DNA repair at such loci[39], one possibility is that in the absence of Trim33, unsupervised DNA replication leads to the accumulation of DNA damage and impairs survival under replicative stress, as described for other conditions that accelerate DNA replication[5].

Loss of Trim33 also counteracts Myc-induced fork slowing, leads to accumulation of DNA damage in cells with high Myc levels, and delays Myc-driven liver tumorigenesis. Ectopic E2f4 expression rescues Myc-induced fork slowing, and either E2f4 or Recql promote the accumulation of DNA damage in Myc-overexpressing cells, suggesting that they underlie the effects of Trim33 on Myc-induced tumorigenesis. We propose that in the presence of Trim33, Myc-induced fork stalling triggers replication checkpoint signaling to facilitate DNA repair, in line with recent observations[24,68]. Suppression of these surveillance mechanisms in Trim33-deficient cells leads to overt DNA damage and delays tumor development, possibly both by cell-intrinsic effects (increased tumor cell death) and via activation of immune responses. Collectively, our data identify a role for the Trim33-E2f4- Recql axis in the cellular response to oncogene- and drug-induced replicative stress and may provide important insights for the development of targeted cancer therapies.

## Methods

### Plasmids

sgRNA and shRNA oligos were cloned into pSpCas9(BB)−2A-Puro (PX459 V2.0, a gift from Feng Zhang; Addgene plasmid #62988) and pLKO1.puro (a gift from Bob Weinberg; Addgene plasmid #845) vectors, respectively. The psPAX2 and pMD2.G vectors were a gift from Didier Trono (Addgene plasmids #12260 and #1225). Transposon and Sleeping Beauty transposase vectors for hydrodynamic tail vein injection were provided by Daniel Dauch (University Hospital Tübingen).

The Recql and E2f4 ORFs were PCR-amplified from mRNA from HCT116 cells and the PCR products were cloned using the HiFi assembly kit (NEB) in pcDNA3, pGEX-4T3 or the pRRL-Hygro vector[69]. Trim33 expression vectors were a gift of Stefano Piccolo (Addgene plasmids # 20902 and 20903). Trim33 mutations (ED887AA, FN1038AA) and E2f4 DNA binding mutation (65RRIYD → AAIAA) were introduced by site-directed mutagenesis. E2f4-GST fusion proteins were generated by cloning the PCR-amplified E2f4 cDNA fragments (1–413, 1–105, 90–200, 300–413) into the pGEX-4T3 vector using the HiFi assembly kit (Cell Signaling). Oligonucleotide sequences are provided in Supplementary Table 2.

### Cell culture

U2OS cells with inducible Myc alleles were provided by Martin Eilers (University of Würzburg); p19/Nras and p19/Nras/Myc cells were provided by Daniel Dauch (University Hospital Tübingen); LentiX cells (a derivative of HEK293T cells) were provided by Michael Hudecek (University Hospital Würzburg). All cell lines were cultured in DMEM medium supplemented with 10% FBS and 1% penicillin–streptomycin. Selection of transfected or transduced cells was performed using puromycin at 1 µg/mL or hygromycin at 500 µg/mL. HU was used at 1 mM as a standard concentration for 2–24 h. For culture under low concentration, HU was used at 0.1 mM for 48 h. Cells were tested for Mycoplasma contamination using the Mycoplasma PCR Detection Kit (Abm). For serum deprivation, p19/Nras cells were allowed to adhere and then cultured in DMEM without FBS for 96 h before release in a complete medium.

For crystal violet staining, cells were fixed in 70% EtOH, air-dried, and stained with crystal violet solution (Sigma) for 20 min at RT, washed with water, and air-dried. The dye was eluted in Methanol: Acetic acid solution and absorbance at 595 nM was quantified using the Tecan plate reader.

### Transfection and lentiviral transduction

Transient transfection for ubiquitin pulldown or immunoprecipitation analysis was performed using polyethylenimine. 24 h post transfection, the medium was replaced and cells were incubated for a further 24 h before harvesting for analysis or antibiotic selection. For siRNA transfection ON-TARGETplus siRNA SMARTpool against mouse E2f4 or Recql or the Non-targeting siRNA Pool (Horizon Discovery, L-054294-00-0005, L-044778-01-0005 D-001810-10-05) were transfected into p19/Nras Trim33-WT and Trim33-KO cells using Dharma-FECT 2 reagent and cells were analyzed 72 h post transfection. For lentivirus production, LentiX cells were co-transfected with the transfer plasmid, packaging, and envelope vectors (psPAX2, pMD2.G) using polyethylenimine. 48 h after transfection, the medium was collected, filtered, and incubated with target cells for 48–72 h in the presence of polybrene before antibiotic selection.

### Immunofluorescence and proximity ligation assays

Cells were cultured on 10 mm glass coverslips, fixed with 1% formaldehyde in PBS for 10 min, and washed three times with PBS. Cells

were permeabilized with PBS/0.2% Triton-X-100 for 10 min and blocked with 5% BSA in PBS/0.2% TX-100 for 10 min. Cells were incubated for 2 h with primary antibodies (E2f4−Proteintech 10923-1-AP, 1:100; Trim33−Sigma HPA004345, 1:100), and secondary antibodies (Cell Signaling anti-rabbit IgG, Alexa Fluor® 555 Conjugate, 1:100) in the blocking solution at room temperature. Coverslips were mounted using the hard set mounting medium with DAPI (Vector Laboratories).

The PLA assay was performed using the Duolink Proximity Ligation Assay kit (Sigma) according to the manufacturer's protocol and antibody pairs as indicated. For EdU labeling and biotin-Recql PLA, cells were treated with HU for 4 h, washed twice with PBS, and released into a fresh medium labeled with 50 μM EdU for 10 min. Cells were fixed with formaldehyde, permeabilized, and incubated with primary antibodies as detailed above and the click reaction was performed in the presence of 0.4 μM biotin-azide, 2 mM CuSO$_4$, and 20 mM Na Ascorbate for 1 h. The PLA assays were then performed with antibodies to biotin and Recql.

The following antibodies were used for PLA assays, all at 1:100 dilution: Recql (Santa-Cruz sc-166388), Recql (Abcam ab151501); HA-tag (Cell Signaling C29F4), RNAPII (Cell Signaling 14958), Mcm2 (Cell Signaling D7G11), RNAPII (Cell Signaling 14958), pS5-RNAPII (Cell Signaling 13523), pS2-RNAPII (Abcam ab5095), biotin (Santa Cruz sc-101339).

## Immunoprecipitation and immunoblotting

Cells were harvested by scraping, washed with PBS, and lysed in TNT-150 lysis buffer (25 mM Tris-HCl pH 8.0, 1% Triton X-100, 150 mM NaCl, supplemented with protease and phosphatase inhibitors), for 30 min on ice. Lysates were cleared by centrifugation and incubated with 2 μg of specific antibody or an isotype control overnight at 4 °C on a rotating wheel. Antibody complexes were collected on protein G agarose beads at 4 °C for 4 h. The beads were washed twice with the TNT-150 lysis buffer for 10 min on ice and captured proteins were denatured in the Laemmli sample buffer at 95 °C for 10 min before immunoblotting analysis.

For immunoprecipitation from formaldehyde-fixed cells, cells were crosslinked with 0.2% formaldehyde in PBS and quenched with 0.2 M Glycine for 5 min. Cells were sonicated in the TNT lysis buffer with 300 mM NaCl and 0.1% SDS, clarified by centrifugation, and immunoprecipitation reactions were performed as above.

Denatured proteins were separated on 10% polyacrylamide Bis-Tris gels using MOPS running buffer (50 mM MOPS, 50 mM Tris, 1 mM EDTA, 0.5% SDS, 5 mM Sodium bisulfite). Proteins were transferred onto PVDF membranes using a semi-dry transfer method in Tris-Glycine buffer (25 mM Tris, 192 mM Glycine, 10% Methanol). For immunoblotting, the primary antibodies were used at 1:1000 dilution and secondary antibodies at 1:5000; the complete list of antibodies with catalog numbers is provided in Supplementary Table 1. Immunoblots were imaged using chemiluminescence reagents (Millipore) on the Bio-Rad Gel-Doc instrument with the ImageLab v5.0 software.

## Ubiquitin pulldown assays

Cells were transfected with vectors encoding 6xHis-tagged ubiquitin, FLAG-tagged Trim33 variants, and HA-tagged E2f4 or Myc using polyethylenimine. 48 h post transfection cells were harvested and lysed in Urea Buffer (8 M Urea/25 mM Imidazole/1% Triton X-100/350 mM NaCl in PBS) and briefly sonicated to shear genomic DNA. Cleared lysates were incubated with Ni-NTA beads (Cube Biotech) overnight to capture ubiquitinated proteins. Beads were washed three times with the lysis buffer and precipitated proteins were denatured in the Laemmli sample buffer at 95 °C for 10 min before separation on Sodium dodecyl-sulfate−polyacrylamide gel electrophoresis (SDS-PAGE) and immunoblotting.

## GST-pulldown assays

Vectors for expression of E2f4-GST fusion proteins or the empty pGEX-4T3 vector were transformed into BL21 DE3 Rosetta cells (Millipore) and protein expression was induced by IPTG. Bacterial pellets were resuspended in PBS with 0.5% TX-100, supplemented with protease and phosphatase inhibitor cocktail, and lyzed by sonication. Clarified bacterial lysates were incubated with glutathione agarose beads to capture GST fusion proteins. The beads were washed three times with the TNT-300 buffer and incubated with mammalian cell lysates for 4 h. Glutathione beads with recovered protein complexes were washed three times with the TNT-300 buffer; proteins were denatured in the Laemmli sample buffer at 95 °C for 10 min before separation on SDS-PAGE and immunoblotting analysis.

## Nascent chromatin capture

The assay was performed as described previously[43] with slight modifications. Cells were treated with HU for 4 h and then incubated for 5 min in a hypotonic buffer (50 mM KCl, 10 mM HEPES pH 8.0) containing biotin-dUTP followed by a 10-min incubation in fresh DMEM medium supplemented with biotin-dUTP. Cells were fixed in 0.2% formaldehyde for 5 min and cross-linking was stopped by adding glycine to a final concentration of 0.2 M for 5 min. Cells were resuspended in the TNT-300 buffer and sheared by sonication with the Hielscher UP200St instrument (2 mm probe, 100% pulse, 30% amplitude, 40 s on, 20 s off for 10 min). The clarified lysates were precipitated using neutravidin agarose overnight, the precipitates were washed with lysis buffer and boiled in the Laemmli sample buffer before immunoblotting analysis.

## FACS analysis

Cells cultured on 10 cm dishes were harvested by trypsinization at 80 % confluency. The cell pellet was resuspended in 200 μL PBS and 800 μL 100% EtOH was added dropwise while vortexing for a final concentration of 80 % EtOH. Cells were incubated at 4 °C overnight, spun for 5 min at 2000 rpm, and washed once with PBS. The cell pellet was resuspended in 200 μL PBS containing 0.5 μL RNAseA and incubated for 20 min on the rotating wheel at room temperature. 10 μL of 1 mg/mL propidium iodide solution was added and cells were incubated for 30 min on the rotating wheel at room temperature in the dark. Data acquisition was performed on the BD LSR Fortessa instrument using the FACSDiva software v8.0. For data analysis, the Flowing software v2.5.1 was used.

## Mass spectrometry

For mass spectrometry analysis of E2f4 interactions, three independent immunoprecipitation reactions with E2f4 antibodies (Proteintech 10923-1-AP) and Isotype Control rabbit IgG (Cell Signaling 3900) from the TNT-150 lysates of Trim33-WT and Trim33-KO p19/Nras cells were performed. The MS analysis was carried out on an Ultimate3000 RSLC system coupled to an Orbitrap Fusion Tribrid mass spectrometer operated by Xcalibur 4.1 SP1 Build 4.1.50. Tryptic peptides were separated on the analytical column (75 μm i.d. × 25 cm, Acclaim Pep-Map RSLC C18, 2 μm, 100 Å; Thermo Fisher Scientific) by a linear gradient from 2% to 30% of buffer B (80% acetonitrile and 0.08% formic acid in HPLC-grade water) in buffer A (2% acetonitrile and 0.1% formic acid in HPLC-grade water) followed by a short gradient of 30% to 95% buffer B. Total gradients lasted 120 min at a flow rate of 300 nL/min. Full MS spectra were acquired with a scan range of 335−1500 $m/z$ and a resolution of 120,000 at $m/z = 200$. MS/MS acquisition was performed in top speed mode with 3 s cycle time with a maximum injection time of 50 ms. The AGC target was set to 400,000 and the isolation window was 1.6 $m/z$. Positive ions with charge states 2−7 were sequentially fragmented by higher energy collisional dissociation.

## Mass spectrometry data analysis

MS Data analysis was performed using the MaxQuant software (version 1.6.17.0) and the integrated Andromeda search engine. Peptide search was performed against the mouse SwissProt database (Release 2020_10) with a peptide tolerance of 20 ppm for the first search and 5 ppm for the main search. Unique and razor peptides with a minimum length of 6 amino acids were used for LFQ quantification. The match-between run option was enabled with a match time window of 0.7 min and an alignment time window of 20 min. For Maxquant processing, protein and peptide spectral match FDR was set to 0.01. Further analysis was performed using Perseus (version 1.6.14.0). Protein groups with at least 2 out of 3 hits in one or more bait groups were considered and missing values were imputed from a normal distribution. Putative interaction partners were defined using two-sample t-tests.

Mass spectrometry data have been deposited in the PRIDE database under PXD037854. All technical details and parameters used for LC-MS sample processing and experimental details of biological replicates and data analysis parameters are provided at the above project link from Proteomexchange.

## RNA sequencing

RNA was isolated using RNeasy Kit (Qiagen). RNA was purified and washed via precipitation with 80% ethanol and quantified using NanoDrop 1000 (Peqlab). For reverse transcription and NGS library preparation, the NEBNext Ultra RNA Library Prep Kit for Illumina (E7530) and NEBNext Multiplex Oligos for Illumina, Dual Index Primer Set 1 (E7600) were used according to the manufacturer's instructions. DNA fragments and index PCR products were purified with HighPrep (MagBio) beads. The sequencing library concentration was determined using the Quant-iT PicoGreen dsDNA Assay-Kit (Thermo).

## ChIP-sequencing and Cut&Run

ChIP was performed as described previously[79] with minor modifications. Briefly, p19/NRas cells were fixed with 1% formaldehyde for 5 min and quenched for 5 min with 0.2 M glycine. Cells were washed with PBS and resuspended in ChIP lysis buffer 1 (25 mM Tris-HCl pH 8.0, 1% Triton X-100/200 mM NaCl/protease & phosphatase inhibitor cocktail) and incubated 30 min on ice. Pelleted nuclei were resuspended in MNase buffer (20 mM Tris pH7.5, 1% TX-100, 5 mM $MgCl_2$) and treated with micrococcal nuclease (NEB) for 3 min. ChIP lysis buffer 2 (25 mM Tris-HCl pH 8.0, 1% Triton X-100, 300 mM NaCl, 0.1% SDS, 10 mM EDTA to 10 mM) was then added and chromatin was fragmented using Hielscher sonifier for 10 min (30% power, 100 amplitude, 40 s on, 20 s off). Cleared lysates were incubated with E2f4 antibody (10923-1-AP, Proteintech) overnight at 4 °C on a rotating wheel. Antibodies were collected using magnetic protein A beads followed by proteinase K treatment and reversal of crosslinks at 65 °C overnight. The DNA was purified by phenol/chloroform extraction and precipitated with ethanol. Sequencing libraries were prepared using the NEB Ultra DNA library Prep kit and the NEBNext Multiplex Oligos for Illumina, Dual Index Primer Set 1. The samples were sequenced on the Illumina Nextseq500 instrument (Nextseq Control Software v2).

Cut&Run was performed on 5*10^5 formaldehyde-fixed p19/Nras cells expressing HA-Recql or HA-E2f4 variants using the CUT&RUN Assay kit (NEB #86652) with 1 μg of antibodies to HA-tag (6E2, Cell signaling) or Recql (ab151501, Abcam) according to the manufacturer's instructions. The released DNA fragments were incubated with proteinase K for 4 h at 65 °C, extracted with phenol-chloroform (Roth), and precipitated with 2.5 volumes of ethanol overnight at −20 °C. DNA was recovered by centrifugation and used for sequencing library preparation using the NEB Ultra II DNA Library Prep kit.

## DSB capture and sequencing

Cells were fixed with formaldehyde and permeabilized with 0.2% Triton X-100 in PBS. DNA end repair reaction was performed by incubation with E. Coli DNA Polymerase I Klenow fragment for 30 min at 25 °C. Cells were washed and an A-tailing reaction was performed in dA-tailing buffer (NEB) with Klenow exo- for 30 min at 37 °C. dA-tailed DNA fragments were ligated to a biotinylated NEB adapter at 16 °C overnight. Cells were lysed in 20 mM Tris pH 8.5, 300 mM NaCl, 0.2% SDS and treated with proteinase K, followed by sonication, phenol-chloroform extraction, and ethanol purification of DNA. Biotinylated DNA fragments were captured on Streptavidin beads and used for library preparation with the NEB Ultra II DNA library end prep kit.

## NGS data analysis

All sequencing data were mapped to mouse genome assembly mm9 using STAR (version 2.5.4). Differential gene expression analysis was performed using EdgeR (version 3.26.8). ChIP, Cut&Run, and DSB-capture data were analyzed using Homer (version 4.10.3). Sequencing data have been deposited in NCBI's Gene Expression Omnibus and are accessible through GEO Series accession number GSE217524.

## Neutral comet assay

Cells were treated as indicated and collected by trypsinization. Cell pellets were washed with PBS and resuspended in 200 μL of 0.7% LMP agarose, 65 μL of the mixture was dropped on a glass slide pre-coated with 0.8% regular agarose and covered by a coverslip. After solidification, another 80 μL of 0.7% LMP agarose was added to cover the cell layer. Cells were lysed in lysis solution (2.5 M NaCl, 0.1 M EDTA, 10 mM Tris pH 10, 1% N-lauroylsarcosine, 0.5% Triton X-100, 10% DMSO) in dark at 4 °C overnight and electrophoresis was preceded in TAE buffer with 0.5 V/cm for 1 h. Cells were fixed with 100% ethanol and stained with ethidium bromide (2 μg/mL in water) for microscopy.

## DNA fiber assay

Cells were incubated for 20 min with 25 μM IdU in complete DMEM, washed with PBS, and incubated for 20 min with 250 μM CldU in complete DMEM. For U2OS cells, expressing shCtrl and shTrim33, the incubation was extended to 40 min for both dNTPs. Cells were harvested by trypsinization, resuspended in PBS, and transferred onto a glass slide. Lysis solution (100 mM Tris pH 8.0, 20 mM EDTA, 1% SDS) was added and slides were tilted to allow DNA spreading. DNA was fixed with Methanol: Acetic Acid (3:1) before incubation with 2.5 N HCl for 90 min. The slides were blocked and the DNA fibers were stained with antibodies against IdU (Cell Signaling 5292 + anti-mouse Alexa555) and CldU (Abcam ab6326 + anti-rat Alexa 488). Image acquisition was performed with Olympus DP80 camera and Olympus BX63 microscope using the cellSens software. Fiber length was measured using ImageJ software version 1.53f. For DNA fiber data analysis, we used the robust non-parametric Kruskal–Wallis test, since the assumptions of normal population distribution and homoscedasticity, which are required for the parametric ANOVA test, in many cases are not fulfilled.

## Mouse model of liver tumorigenesis

The animal experiments were approved by the regional government of Tübingen (authorization number M09/20G). Wild-type mice C57BL/6 mice were purchased from Charles River (Sulzfeld) and housed at 21–22 °C ambient temperature, 40–60% humidity, and a 12/12 h light/dark cycle, in accordance with the institutional guidelines of the University Hospital Tübingen.

DNA for hydrodynamic tail vein injection was prepared using the Qiagen EndoFreeMaxi Kit. DNA was diluted in saline solution at a final volume of 10% of body weight. 4- to 6-week-old mice (7 mice/group) were injected with 25 μg of the transposon plasmids expressing Myc, Ras, and shRNA and 5 μg of a Sleeping Beauty transposase-expressing vector. Mice were monitored for tumor development by palpation and sacrificed upon reaching a critical tumor burden. Livers were explanted and examined for tumors, photographed under day- and fluorescent light, and fixed in PFA for immunochemistry analysis.

## Immunohistochemistry of murine liver samples

After heat-induced antigen retrieval at pH 6, formalin-fixed and paraffin-embedded tissue sections were incubated for 1 h at room temperature with a TRIM33 antibody (Sigma HPA004345, dilution 1:50), pH2AX antibody (Cell Signalling #9781, dilution 1:100), and a CD3 antibody (Thermo Scientific RM-9107, dilution 1:100), respectively. An anti-rabbit secondary antibody conjugated to AP (Polyview Plus AP (anti-rabbit) reagent, ENZO Life Sciences GmbH, Lörrach, Germany) was applied. The signal was visualized using alkaline phosphatase (Permanent-AP Red Kit, Zytomed Systems, Berlin, Germany) as a chromogen. pH2AX and CD3 stained sections were quantified manually using the Aperio ImageScope—Pathology Slide Viewing Software 12.4.6.

## Reporting summary

Further information on research design is available in the Nature Portfolio Reporting Summary linked to this article.

## Data availability

The NGS data have been deposited in NCBI's Gene Expression Omnibus and are accessible through GEO Series accession number GSE217524. The Mass spectrometry data have been deposited to the ProteomeXchange Consortium via the PRIDE partner repository under the accession code PXD037854. Source data are provided in this paper. All other raw data and materials are available from the corresponding author upon request. Source data are provided in this paper.

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

## Acknowledgements
We thank Kseniya Popova and Veronika Eckel (NCT, Department of Pathology, University Hospital Heidelberg) for expert technical assistance. This work was supported by the German Research Foundation (DFG) grants PO1458-3/2 and 10059-1 (iFIT cluster EXC_2180) to N.P. and an institutional grant from the German Cancer Aid to M.F.

## Author contributions
N.P. conceived the study and designed experiments. V.R., E.E., C.J., M.A.J., J.H., T.P., M.R., and N.P. performed experiments. N.P. and M.F. supervised experiments and interpreted the data. N.P. wrote the paper.

## Funding

## Competing interests
The authors declare no competing interests.
