## [Peer Review File · Nature Communications]

Trim33 masks a non-transcriptional function of E2f4 in replication fork progressionREVIEWER COMMENTS

Reviewer #1 (Remarks to the Author):

There are certain elements of the proposed model, which are very interesting and intriguing. The fact that Trim33 is an ubiquitin ligase for E2F4, for example, is interesting and potentially important finding. However, the key claim that this interaction mediates a non-transcriptional function of E2F4 is not sufficiently supported by the available data and I think the key data of the manuscript argue against this claim.

One very puzzling finding is that the reduction of DNA synthesis caused by HO-urea is reverted by E2F4 and Trim33 deletion. HO-urea reduces levels of desoxyribonucleotides, so if DNA synthesis is relatively unimpaired, the cells must have sufficient stores of them. This indicates that under the conditions where this occurs (high rate of DNA synthesis in the presence of HU), there is a lot of biosynthesis, potentially because a biosynthetic gene (e.g. RRM2?) genes is highly induced by E2F4. So the argument that these findings indicate a direct role in the relieve of replication stress appears vague and the mechanism that underlie the data shown panels 1E and 1F, for example, remain unclear.

Furthermore, while the data showing that Trim33 regulates E2F4 levels are clear, all co-immunoprecipitation assays showing that E2F4 interacts with REQL1 are unconvincing. The bands in panels 3E and 4B,C are too weak and too close to background to be convincing. In combination with the very nice PLA assays, this argues that REQL1 is in proximity with E2F4, but does not interact – which is counter the main argument.

And finally, the authors show unequivocally and very nicely that Trim33 regulates E2f4 transcriptional function(Figure 3c)

Additional comments

The key Figure 2A is unclear. It is uncertain whether this type of analysis can really distinguish one of the E2F isoforms with sufficient accuracy from the other family members and the fact that RNA Polymerase is enriched on place 2 of the list raises concerns that this is mainly a list of active promoters.

In Figure 5C, the RECQL binding is clearly not centered around the TSS, but upstream. This is an interesting observation, since it could point to conflicts with the replication forks (since origins are upstream of promoters).

The CHX time course in Figure 2B is unusual and has a single point where there is a difference. This needs to be supported by statistics from multiple experiments.

Reviewer #2 (Remarks to the Author):

Highlights:

- Upon replicative stress induced by HU or Myc, E2f4 recruits the DNA helicase Recql to accelerate progression of DNA replication forks.
- In the absence of external stress, the ubiquitin ligase Trim33 targets E2f4 for degradation and diminishes its interaction with Recql as well as its binding to genomic sites.
- Replicative stress reduces Trim33-dependent ubiquitination of E2f4, resulting in genome-wide recruitment of Recql and enhancement of DNA synthesis.
- Depletion of Trim33 in Myc-overexpressing cells exacerbates replication-associated DNA damage and

delays Myc-driven tumorigenesis.

- The Trim33-E2f4-Recq1 axis regulates the progression of DNA replication forks to maintain genomic integrity.

General comments

1. In general, this is a very insightful study, adding a new dimension to the role of E2F4 in S phase. It has long been known for inhibiting G1-S transition, but a direct impact on replication fork progression is thought-provoking and surprising. Moreover, the functional interaction of Trim33 and E2F4 is of high interest to a general readership.
2. The idea of the paper is that removing Trim33 enhances E2F4 recruitment to promoter DNA, and that this would enable enhanced Recq1 binding to chromatin. Along this line, removing Trim33 not only enhances E2F4 recruitment, but also Recq1 recruitment (Fig. 4). But what if Trim33 AND E2F4 are removed, at least temporarily by knockdown? Would this reduce the amount of Recq1 on chromatin again? Such an experiment would corroborate a causal role of E2F4 in Recq1 recruitment.
3. What is the role of Trim33 in the regulation of the cell cycle, and in particular in the transition from G1 to S and from G2 to M? Those are the "classical" roles of E2F4 and the associated Rb-like proteins. Does Trim33 compromise E2F4-mediated cell cycle arrest? Even if the answer is no, it would be an obvious question to address, and even "negative" results on this would be worth reporting.
4. Along the same line: Could the authors provide any data on a possible competition between the pocket proteins and Recq1 binding, especially since the carboxyterminal domains of E2F4 bind both? For instance, would depleting the pocket proteins affect the co-immunoprecipitation of E2f4 and Recq1?
5. Replication stress is known to enhance the (innate or T-cell-mediated) immune response in tumors. It would be helpful if the authors could at least begin to test this by staining the tumors from their animal models for corresponding markers.
6. The authors propose that Trim33 associates with E2f4 on chromatin, based on the observation that E2f4 domains that mediate DNA binding or histone recognition are required for the interaction (139). To corroborate this statement, the Co-IP experiments should be performed with and without nuclease (e.g., benzonase) treatment of cell lysates, in order to see if the interaction of the two factors depends on the presence of DNA. Moreover, the interaction should be analyzed using recombinantly expressed, purified proteins.
7. Why are different statistical tests used to analyze the DNA fiber assays? It would be more accurate if the authors could add a "Statistical analysis" paragraph in the method section to describe the statistical tests more carefully. Moreover, the number of fibers analyzed in the fiber assays is quite low, and these assays are notorious for variability. Could the authors increase the number of analyzed fibers to a minimum of 100?
8. In Figure S1A, Trim33 KD by shRNAs strongly reduces the endogenous levels of Myc. Still, upon Myc overexpression, the levels of Myc appear to be slightly reduced by Trim33 KD. Could the authors exclude that Myc-induced fork slowing, along with the reduction of replication stress markers, is less prominent upon Trim33 KD because of the reduction in Myc levels? A reduction of Myc protein levels upon Trim33 KD is also visible in U2OS cells (Fig. S1C).
9. DNA replication fork progression was mostly tested after incubation with HU. Could the authors measure DNA elongation upon treatment with other agents causing replication stress? E.g., topoisomerase inhibition and/or Pol alpha inhibition? This would render their conclusions more general.
10. Similarly, could the authors elaborate more on the fact that RecqI is recruited more prominently in the context of the HU treatment (replication stress)? Upon replication stress, the recruitment of RecqI could cause untimely restart of stalled replication forks, leading to DNA damage accumulation. Could the authors comment on the biological role of RecqI recruitment upon replication stress? At present, the observations might reflect fork stalling upon nucleotide deprivation, rather than actual DNA damage that forms an obstacle to DNA replication fork progression. Could the authors therefore investigate further whether RecqI recruitment is enhanced upon treatment with other replication stress-inducing agents, such as inhibitors of topoisomerases, Pol alpha, ATR, Chk1, or nucleoside analogues?

Comments to figures

11. Figure 1C: there is no statistical data displayed between KO ctrl and KO Myc.
12. Figure S2C only reports representative images of the IF staining. The authors should add a quantification including at least 100 cells per sample. Could the authors also show images of the DAPI and E2f4 signal as merge? This would help to identify the intracellular the localization of the signal.
13. Figure 2B: the authors should quantify and report the intensity of the immunostaining for the three biological replicates and calculate statistical significances.
14. Figure 3E displays a weak signal for Recq1 that is barely over the background signal. Could the authors provide more biological replicates and a quantification?
15. Figure 5A: how many biological replicates have been performed for this PLA?
16. Figure 7: the number of animals needs to be reported
17. Figure 4A: The description of the last two samples is inaccurate. Both are named WT Mcm2.

Minor comments

18. The intensity of the signal in the IPs should be enhanced. The authors could show additional, longer exposure time.
19. 232: reference to figure is imprecise.
20. Please add the abbreviations with explanations to the figure legends.
21. Could the authors specify which gene is WT or KO in all the figures, not only in the figure legend?
22. Figure 4G: could the authors state the cells used in each figure legend?
23. The abbreviation for transcription-replication conflicts (TRCs) is introduced three times.
24. Figure 1C, and all the fiber assays scheme should report the time of labeling with CldU and IdU.
25. Figure S1D (D) may refer to the wrong figure, please check.
26. Figure 1E: the scheme representing the HU treatment should be expanded to show the incubation time.

Reviewer #3 (Remarks to the Author):

In this study Rousseau et al., show that the Trim33 ubiquitin ligase limits replication-associated DNA damage induced by genotoxic agents and MYC oncogene. Mechanistically, authors show that Trim33 regulates E2f4 transcription factor stability and restricts interactions of E2f4 with Recq1 helicase to control progression of DNA replication forks under replicative stress. Moreover, authors show that deletion of Trim33 triggers ectopic recruitment of E2F4 and Recq1 to chromatin, leading to rapid DNA replication, defective checkpoint signalling and DNA damage. Finally, by using an autochthonous model of liver cancer, authors show that Trim33 deficiency impairs survival under replicative stress and delays Myc-dependent liver tumorigenesis. The manuscript is interesting, and is overall properly done and well written, however, before publication can be recommended, several points need to be addressed in order to support the main conclusions of the work. With this aim, new experiments are suggested. Also, some points should be clarified or corrected.

Specific points that deserve additional attention are outlined below:

- Although loss of Trim33 led to the deregulation of DNA replication and cell cycle genes, knockout of Trim33 did not significantly affect replication fork progression in unchallenged conditions. How are the cell cycle profile and DNA replication status of these cells? Moreover, the DNA damage present in these cells in unchallenged conditions should be assessed, since gH2AX levels were only measured upon HU treatment (Figure 6).
- According to Figure 1, authors conclude that "depletion of Trim33 suppresses Myc-induced replicative stress and the accompanying activation of DDR signalling". Strikingly, in the last section of the results authors state that "Loss of Trim33 exacerbates Myc-induced DNA damage". These seem opposing messages and need clarification.

- Related to the previous one, data shows that TRIM33 loss leads to increased gH2AX levels but decreased pKAP1 and pRPA DNA damage markers, as well as 53BP1 bodies. It would be necessary to discuss these conflicting data. Authors suggest that checkpoint signaling might be compromised upon Trim33 loss. Is anything known in this line?

- In Supplementary Figure 1D, DNA fiber assay is performed in U2OS cells with inducible expression of MYC, and authors conclude that TRIM33 loss rescues MYC induced fork slowing. However, this rescue is not very clear with knockout number 5. These analyses could benefit from additional fiber measurement, the measurement of 100 fibers in this kind of assay is usually preferred. Moreover, how were these U2OS knockout cell lines generated? In order to avoid long term effects of TRIM33 loss, I would recommend to perform acute depletion of TRIM33 in these assays. Moreover, if shRNAs were used, I would suggest to use "KD" instead of "KO" in Supplementary Figure 1C and D.

- Authors found that deletion of Trim33 stabilized E2F4 transcription factor, and their interaction might involve the DNA binding domain of E2F4. Considering that the DBD of E2F transcription factors is conserved among the different members of the E2F family of transcription factors, it would be interesting to know whether this function of Trim33 is specific for E2F4, or whether it regulates the stability of other E2F factors.

- Regarding the function of Trim33 in regulating E2F4 stability, the immunofluorescence images of E2F4 upon Trim33 deletion are not very clear. Quantification of the images would be helpful, as well as choosing new images that better represent the result. On the other hand, why are not E2F4 protein levels increased in steady state conditions upon Trim33 loss in Figure 2D? Moreover, since authors claim that DNA damage or MYC overexpression overrules Trim33 mediated E2F4 levels, are E2F4 levels increased under these conditions? Authors suggest that E2F4-Trim33 interaction is decreased upon HU, but data is not very convincing (Figure 5E). This is an important point, authors should try other approaches to make this point stronger.

- Regarding E2F4 immunoprecipitations, there are several issues that should be considered. In Figure 2D, Trim33 band in E2F4 IP sample is very hard to see, even in the MG132 condition. Also, Co-IP analyses between E2F4 and Recq1 are not very clear. Performing these IP-s with exogenously expressed proteins or crosslinked samples would strengthen the data (as done in Figure 4B). Which band corresponds to E2F4 in Figure 2D? Regarding E2F4 mutants (Figure 2F), authors claim that E2F4-KR and E2F4-deltaC enhance the interaction with Trim33, however, Trim33 levels are also increased in the input samples. The decreased interaction between E2F4 and Trim33 upon HU treatment in Figure 5E, is hardly appreciated. Overall, IP data are not very strong and authors should improve them.

- According to Figure 2G, E2f4 depletion slowed down DNA replication forks during recovery from HU treatment in Trim33-KO cells, and expression of E2f4 variants that lack DNA binding domain or the C-terminal region, decelerated DNA replication and equalized fork progression in Trim33-WT and Trim33-KO cells (Figure 2H). Were experiments in Figure 2H performed without HU? If so, how do the authors explain the effect of these mutants considering that E2F4 depletion did not have any effect in unchallenged conditions? These data need better explanation and discussion.

- Authors found that HU induced a robust recruitment of Recq1 to chromatin in Trim33-WT cells, which was especially prominent at E2f4-bound sites. It would be important to assess whether the increase of Recq1 binding upon HU is the consequence of an increased E2F4 recruitment to these regions. Moreover, according to ChIP-Seq and RNA-Seq data, for most E2f4 binding sites, enhanced E2f4 recruitment upon Trim33 loss was not associated with altered gene expression. It would be interesting to assess if this is also the case upon HU treatment or MYC overexpression, at least in selected target genes. This may reinforce the idea that E2F4 has a non-transcriptional role in DNA damage conditions.

- Regarding the analysis of transcription-replication conflicts in Figure 5G, H, authors should clarify why collisions for pS5-RNAPII were reduced upon HU treatment, whereas collisions for total RNAPII did not decrease upon HU in Trim33-WT cells. On the contrary, the interaction with PCNA of both, pS5-RNAPII and total RNAPII, were reduced upon Trim33 loss. Moreover, authors should check how Recq1 interaction with RNAPII and Mcm2 is affected upon HU.

- In Figure 6, authors used the E2f4- Δ C (which does not recruit Recq1), mutant and observed decreased pHA2X levels in Trim33-KO cells following release from HU, and suggest that recruitment of Recq1 may underlie accumulation of DNA damage in the absence of Trim33. However, authors should consider that E2f4- Δ C mutant is not only deficient for Recq1 binding. These analyses might be performed together with Recq1 depletion, in order to strengthen the role of Recq1 in the increased DNA damage observed in Trim33 deficient cells. Moreover, in order to strengthen that Trim33 loss promotes DNA damage and impairs cell growth via E2F4-Recq1 recruitment, colony survival assays should be performed upon E2F4 and Recq1 loss.

- In Figure 4F, it would be important to include the fiber data of samples transfected with a scramble siRNA.

- If the E2f4-Recq1 interaction does not require the DNA-binding domain (Figure 3G), why do the authors suggest that E2f4 may facilitate Recq1 localization to chromatin?

- Molecular weights should be included in all the blots presented in the manuscript. This is especially important when analysing protein ubiquitylation.

- For the RNASeq data, it is important to establish what criteria was used regarding the differential expression analysis.

- For all the cell treatments performed, it is important to include the treatment conditions in the figure legends.

“Trim33 masks a non-transcriptional function of E2f4 in replication fork progression”
Rousseau et al.

Point-by-point response to reviewer comments

Reviewer #1 (Remarks to the Author):

There are certain elements of the proposed model, which are very interesting and intriguing. The fact that Trim33 is an ubiquitin ligase for E2F4, for example, is interesting and potentially important finding. However, the key claim that this interaction mediates a non-transcriptional function of E2F4 is not sufficiently supported by the available data and I think the key data of the manuscript argue against this claim.

We are happy that the reviewer finds our work interesting and thank the reviewer for constructive criticism and suggestions. In the revised version of the manuscript, we provided further evidence that supports our original model of non-transcriptional role of E2f4 in DNA replication. In particular, we assessed recruitment of E2f4 to chromatin and gene expression profiles in HU-treated cells. We show that E2f4 is recruited to chromatin in response to hydroxyurea (HU) (Figure 5h,i), which is not accompanied by transcription changes of E2f4 target genes (Supplementary Figure 5g,h). In contrast, E2f4 binding correlates with Recq1 recruitment to chromatin (Figure 5i). Furthermore, we show that E2f4 is required for Recq1 binding to chromatin in Trim33-KO cells (Figure 4d,e,f,g) and upon HU treatment (Supplementary Figure 5i).

In addition to the arguments in the original manuscript (pages 4 and 5), we believe that the following findings - 1) Recq1 protein levels are not affected by loss of Trim33, 2) E2f4 is required for Recq1 recruitment to chromatin in Trim33-KO cells and under HU (Figure 4d,e,f,g; Supplementary Figure 5i), 3) recruitment of E2f4 to chromatin upon HU treatment is not accompanied by altered expression of E2f4 target genes, but is required for acceleration of replication forks under stress (Figure 5h,i; Supplementary Figure 5j), 4) both E2f4 and Recq1 are required for accelerated DNA replication and for accumulation of DNA damage in Trim33-deficient cells upon replicative stress (Figure 6d,e; Figure 7b,c) - strongly argue that E2f4 regulates DNA replication under stress independently of target gene expression.

One very puzzling finding is that the reduction of DNA synthesis caused by HO-urea is reverted by E2F4 and Trim33 deletion. HO-urea reduces levels of desoxyribonucleotides, so if DNA synthesis is relatively unimpaired, the cells must have sufficient stores of them. This indicates that under the conditions where this occurs (high rate of DNA synthesis in the presence of HU), there is a lot of biosynthesis, potentially because a biosynthetic gene (e.g. RRM2?) genes is highly induced by E2F4. So the argument that these findings indicate a direct role in the relieve of replication stress appears vague and the mechanism that underlie the data shown panels 1E and 1F, for example, remain unclear.

We assume that the reviewer was under the impression that the fiber assays were performed in the presence of hydroxyurea, perhaps due to insufficient explanation or labeling. However, our experiments were performed after the hydroxyurea was removed from cell culture media followed by the addition of labeled dNTPs - the standard conditions to assess the restart and progression of stalled replication forks. As a control, we now performed the assay with HU present in cell culture media. Under these conditions, DNA replication is completely abolished in both Trim33-WT and Trim33-KO cells (please see Figure 1a below). We also compared protein levels of Rrm2, which could accelerate DNA synthesis after HU release and found no significant difference in Trim33-WT and Trim33-KO p19Nras cells (Supplementary Figure 3b) and in shCtrl and shTrim33 U2OS cells (Figure 1b below).

Figure 1. a) DNA fiber assays in p19/Nras Trim33-WT and Trim33-KO cells, labelled under normal growth conditions or in the presence of 1mM HU. **b)** Expression of RRM2 in shCtrl and shTRIM33 U2OS cells.

Furthermore, while the data showing that Trim33 regulates E2F4 levels are clear, all co-immunoprecipitation assays showing that E2F4 interacts with REQL1 are unconvincing. The bands in panels 3E and 4B,C are too weak and too close to background to be convincing. In combination with the very nice PLA assays, this argues that REQL1 is in proximity with E2F4, but does not interact – which is counter the main argument.

We repeated co-immunoprecipitation experiments with E2f4 antibodies in benzonase-treated lysates to release chromatin-associated complexes and obtained stronger data in Trim33-KO p19Nras cells (Figure 3f) and in U2OS cells following Myc induction (Supplementary Figure 5f). The interaction is also readily detectable by immunoprecipitation of endogenous Recql from formaldehyde-crosslinked Trim33-KO p19/Nras cells, stably expressing HA-tagged E2f4 variants (Supplementary Figure 3e). We also used pulldown assays with GST-fusion proteins, containing E2f4 fragments and found that endogenous Recql can be captured with the C-terminal fragment of E2f4 that includes the transactivation domain, supporting the model that the two proteins interact directly (Figure 3f).

And finally, the authors show unequivocally and very nicely that Trim33 regulates E2f4 transcriptional function(Figure 3c)

We agree that Trim33 controls transcriptional activity of E2f4 - this is how we initially identified the link between the two proteins. Notably, the correlation between E2f4 binding and changes in gene expression is obvious only for repressed genes (Figure 3c), whereas replication-associated genes are virtually all upregulated by Trim33 deletion (Supplementary Figure 3a).

Additional comments

The key Figure 2A is unclear. It is uncertain whether this type of analysis can really distinguish one of the E2F isoforms with sufficient accuracy from the other family members and the fact that RNA Polymerase is enriched on place 2 of the list raises concerns that this is mainly a list of active promoters.

Obviously, such analysis will yield a list of active promoters with a certain level of specificity and RNAPII is likely to be an enriched factor for many conditions. For example, analysis of HU-induced transcriptional changes also identified RNAPII but not E2f4 (Supplementary Figure 5g). We agree that such an analysis alone would not be able to discriminate between different E2f family members and initially we focused on E2f4 as a candidate factor. Additional assays showed that Trim33 does not recruit other analyzed E2F family proteins nor regulate their turnover (Supplementary Figure 2f,g).

In Figure 5C, the RECQL binding is clearly not centered around the TSS, but upstream. This is an interesting observation, since it could point to conflicts with the replication forks (since origins are upstream of promoters).

We agree that the Recql binding profile is interesting and could indicate the binding to both promoters and proximally-stalled replication forks. This is in line with previous studies that show the role of Recql in facilitating the restart of stalled replication forks and in the resolution of transcription-replication conflicts (TRCs) (Benedict et al, 2020; Chappidi et al, 2020) and with our observation that Trim33 knockout affects the incidence of TRCs (Figure 5j). This binding pattern is also obvious in Cut&Run experiments with endogenous Recql shown in Figure 4g.

The CHX time course in Figure 2B is unusual and has a single point where there is a difference. This needs to be supported by statistics from multiple experiments.

We provide a different exposure of this immunoblot, in which a gradual decrease of E2f4 protein is clearly visible in Trim33-WT cells after 1 hour of CHX treatment, whereas in Trim33-KO cells

E2f4 is virtually unchanged through the time course. We repeated the assay multiple times with quantification and included these data in Figure 2b.

Reviewer #2 (Remarks to the Author):

Highlights:

- Upon replicative stress induced by HU or Myc, E2f4 recruits the DNA helicase Recq1 to accelerate progression of DNA replication forks.
- In the absence of external stress, the ubiquitin ligase Trim33 targets E2f4 for degradation and diminishes its interaction with Recq1 as well as its binding to genomic sites.
- Replicative stress reduces Trim33-dependent ubiquitination of E2f4, resulting in genome-wide recruitment of Recq1 and enhancement of DNA synthesis.
- Depletion of Trim33 in Myc-overexpressing cells exacerbates replication-associated DNA damage and delays Myc-driven tumorigenesis.
- The Trim33-E2f4-Recq1 axis regulates the progression of DNA replication forks to maintain genomic integrity.

General comments

1. In general, this is a very insightful study, adding a new dimension to the role of E2F4 in S phase. It has long been known for inhibiting G1-S transition, but a direct impact on replication fork progression is thought-provoking and surprising. Moreover, the functional interaction of Trim33 and E2F4 is of high interest to a general readership.

We thank the reviewer for the positive comments and helpful suggestions.

2. The idea of the paper is that removing Trim33 enhances E2F4 recruitment to promoter DNA, and that this would enable enhanced Recq1 binding to chromatin. Along this line, removing Trim33 not only enhances E2F4 recruitment, but also Recq1 recruitment (Fig. 4). But what if Trim33 AND E2F4 are removed, at least temporarily by knockdown? Would this reduce the amount of Recq1 on chromatin again? Such an experiment would corroborate a causal role of E2F4 in Recq1 recruitment.

We performed several experiments to address this point. First, we depleted E2f4 in Trim33-WT and Trim33-KO cells using siRNA followed by Cut&Run experiments with Recq1 antibodies and analyzed enrichment at several E2f4 and Recq1-bound promoters by qPCR. Depletion of E2f4 strongly reduced recruitment of endogenous E2f4 to these sites. These data are now shown in Figure 4e. Second, we assessed genome-wide recruitment of endogenous Recq1 to chromatin in Trim33-KO cells expressing E2f4 variants that do not bind DNA or Recq1 (E2f4- Δ DB or E2f4- Δ C) compared to cells expressing the full-length E2f4. This assay showed a strong reduction in Recq1 recruitment at most of Recq1-bound sites for both E2f4 variants compared to the full-length protein (Figure 4f,g). These results are supported by immunoprecipitation and PLA experiments with

Recql antibodies in Trim33-KO cells transfected with E2f4 siRNA or expressing the E2f4 variants (Figure 4d; Supplementary Figure 4c,d) that demonstrate a requirement for E2f4 in Recql recruitment to RNAPII and replisome proteins - Mcm2 and PCNA.

3. What is the role of Trim33 in the regulation of the cell cycle, and in particular in the transition from G1 to S and from G2 to M? Those are the “classical” roles of E2F4 and the associated Rb-like proteins. Does Trim33 compromise E2F4-mediated cell cycle arrest? Even if the answer is no, it would be an obvious question to address, and even “negative” results on this would be worth reporting.

Knockout of Trim33 did not affect cell proliferation or cell cycle distribution of the unchallenged cells (Supplementary Figure 2a,b). To specifically assess the impact on G1-S progression, we synchronized p19Nras Trim33-WT and Trim33-KO cells by serum deprivation or lovastatin and measured EdU incorporation at different points after release in fresh medium. Deletion of Trim33 did not significantly affect the onset of DNA synthesis in either experiment. The data for serum deprivation and release are shown in Figure 1i.

4. Along the same line: Could the authors provide any data on a possible competition between the pocket proteins and Recq1 binding, especially since the carboxyterminal domains of E2F4 bind both? For instance, would depleting the pocket proteins affect the co-immunoprecipitation of E2f4 and Recq1?

As E2f4 binds to all three pocket proteins, which would need to be simultaneously depleted, we cloned and expressed a cDNA for FLAG-tagged fragment of p107 encompassing the pocket region and immunoprecipitated E2f4 followed by immunoblotting with Recql antibodies. Co-transfection of this cDNA diminished E2f4-Recql interaction, but we were unable to visualize the expression of the p107 fragment (perhaps due to low expression levels) and cannot include these data in the manuscript. However, we believe that this is an interesting point, relevant for the described mechanism, and we discuss this possibility on page 10.

5. Replication stress is known to enhance the (innate or T-cell-mediated) immune response in tumors. It would be helpful if the authors could at least begin to test this by staining the tumors from their animal models for corresponding markers.

We assessed the presence of T-cells in tumors expressing shCtrl and shTrim33 using immunohistochemistry with the anti-CD3 antibody. In line with higher levels of replicative stress and DNA damage in Trim33-deficient cells, shTrim33 significantly increased the number of CD3-positive T cells in liver tumors. These data are shown in Supplementary Figure 7h and described on page 8 of the main text.

6. The authors propose that Trim33 associates with E2f4 on chromatin, based on the observation that E2f4 domains that mediate DNA binding or histone recognition are required for the interaction (139). To corroborate this statement, the Co-IP experiments should be performed with and without nuclease (e.g., benzonase) treatment of cell lysates, in order to see if the interaction of the two

factors depends on the presence of DNA. Moreover, the interaction should be analyzed using recombinantly expressed, purified proteins.

We performed these experiments and found that benzonase treatment increases the level of E2f4 in the lysates and enhances the recovery of E2f4-Trim33 complexes (Supplementary Figure 2i). Together with mutagenesis data, this result suggests that the E2f4-Trim33 complex is preferentially formed on chromatin and involves direct E2f4-Trim33 interaction.

We performed GST pulldown experiments with GST-tagged E2f4 fragments and U2OS cell lysates and found that the N-terminal region of E2f4 that includes the DNA-binding domain is sufficient for interaction with Trim33 (Figure 2h).

7. Why are different statistical tests used to analyze the DNA fiber assays? It would be more accurate if the authors could add a "Statistical analysis" paragraph in the method section to describe the statistical tests more carefully. Moreover, the number of fibers analyzed in the fiber assays is quite low, and these assays are notorious for variability. Could the authors increase the number of analyzed fibers to a minimum of 100?

We now use the Kruskal-Wallis test to determine significance in all DNA fiber assays as a more stringent and robust test with less requirements for the analyzed data structure; we mention this in the DNA fiber paragraph of the Methods section. The number of DNA fibers in each assay was increased to at least 100.

8. In Figure S1A, Trim33 KD by shRNAs strongly reduces the endogenous levels of Myc. Still, upon Myc overexpression, the levels of Myc appear to be slightly reduced by Trim33 KD. Could the authors exclude that Myc-induced fork slowing, along with the reduction of replication stress markers, is less prominent upon Trim33 KD because of the reduction in Myc levels? A reduction of Myc protein levels upon Trim33 KD is also visible in U2OS cells (Fig. S1C).

Trim33 knockout or depletion reduces the protein and mRNA levels of endogenous Myc as evident in RNA-seq and qPCR assays (Supplementary Figure 2a). The apparent decrease in Fig. S1C (now Supplementary Figure 1e) is most likely due to uneven loading of the samples. We found that the relative levels of exogenous Myc in Trim33-WT and Trim33-KO cells in different immunoblotting experiments vary between 1.6 and 0.7, when considering the loading - for example, shRNA-mediated depletion of Trim33 in U2OS cells slightly increases exogenous Myc (Supplementary Figure 1g,h). The effects on endogenous Myc may be caused, for example, by E2f4-mediated repression of the *Myc* gene (Chen et al, 2002). Since deletion of Trim33 accelerates DNA replication during recovery from different treatments, such as HU, aphidicolin and etoposide (Figure 1e, Supplementary Figure 1j), we believe that changes in endogenous Myc are unlikely to mediate the effect of Trim33 loss on DNA replication.

9. DNA replication fork progression was mostly tested after incubation with HU. Could the authors measure DNA elongation upon treatment with other agents causing replication stress? E.g., topoisomerase inhibition and/or Pol alpha inhibition? This would render their conclusions more general.

We now performed the DNA fiber assays in Trim33-WT and Trim33-KO cells after release from the treatment with another replication inhibitor aphidicolin and a topoisomerase II inhibitor etoposide. Knockout of Trim33 accelerated replication fork progression in both cases (Supplementary Figure 1j), indicating that Trim33 generally limits replication fork progression under stress.

10. *Similarly, could the authors elaborate more on the fact that Recql is recruited more prominently in the context of the HU treatment (replication stress)? Upon replication stress, the recruitment of Recql could cause untimely restart of stalled replication forks, leading to DNA damage accumulation. Could the authors comment on the biological role of Recql recruitment upon replication stress? At present, the observations might reflect fork stalling upon nucleotide deprivation, rather than actual DNA damage that forms an obstacle to DNA replication fork progression. Could the authors therefore investigate further whether Recql recruitment is enhanced upon treatment with other replication stress-inducing agents, such as inhibitors of topoisomerases, Pol alpha, ATR, Chk1, or nucleoside analogues?*

To address this point, we immunoprecipitated Recql from crosslinked cells after exposure to HU, etoposide and doxorubicin, which all induce replication checkpoint signaling. Recql association with RNAPII was enhanced in all cases (Supplementary Figure 5e), arguing that recruitment of Recql to chromatin is a general response to checkpoint signaling. We propose that the enhanced Recql recruitment results from inhibition of Trim33-mediated regulation of E2f4. In line with this idea, different genotoxins that activate the checkpoint, including etoposide, doxorubicin, and topotecan, also upregulate E2f4 protein levels (Supplementary Figure 5d).

Comments to figures

11. *Figure 1C: there is no statistical data displayed between KO ctrl and KO Myc.*

We added the significance value for the KO-Ctrl and KO-Myc samples.

12. *Figure S2C only reports representative images of the IF staining. The authors should add a quantification including at least 100 cells per sample. Could the authors also show images of the DAPI and E2f4 signal as merge? This would help to identify the intracellular the localization of the signal.*

We repeated the assay with a minimum of 100 cells per sample, quantified the data and included the merged images for E2f4 and DAPI signals in Supplementary Figure 2e.

13. *Figure 2B: the authors should quantify and report the intensity of the immunostaining for the three biological replicates and calculate statistical significances.*

We repeated the cycloheximide assays and provide the quantification in Supplementary Figure 2b.

14. *Figure 3E displays a weak signal for Recq1 that is barely over the background signal. Could the authors provide more biological replicates and a quantification?*

We repeated the immunoprecipitation experiments and provide a stronger experiment in Figure 3f for p19Nras cells and in Supplementary Figure 5f for U2OS cells. To corroborate this result, we immunoprecipitated endogenous Recq1 in crosslinked Trim33-KO cells expressing HA-tagged E2f4 variants and show this experiment in Supplementary Figure 3e.

15. *Figure 5A: how many biological replicates have been performed for this PLA?*

The experiment was now performed three times with similar results. We updated the figure legend (now Figure 5b) to indicate this fact.

16. *Figure 7: the number of animals needs to be reported*

We included the number of animals (7 mice per group) in the figure legend and in the methods section.

17. *Figure 4A: The description of the last two samples is inaccurate. Both are named WT Mcm2.*

We corrected the labeling. These data are now shown in Supplementary Figure 4a.

Minor comments

18. *The intensity of the signal in the IPs should be enhanced. The authors could show additional, longer exposure time.*

We repeated many of the IP experiments to increase signal to noise ratio, performed new IP experiments to support our model and provided quantification for several key experiments as requested.

19. *232: reference to figure is imprecise.*

We updated and corrected references to all figures.

20. *Please add the abbreviations with explanations to the figure legends.*

We added the abbreviations with explanations to figure legends.

21. *Could the authors specify which gene is WT or KO in all the figures, not only in the figure legend?*

We specified the gene for WT / KO labels in all the figures.

22. *Figure 4G: could the authors state the cells used in each figure legend?*

We specified which cell line was used in each figure legend.

23. *The abbreviation for transcription-replication conflicts (TRCs) is introduced three times.*

We removed the multiple introductions of the TRC abbreviation.

24. *Figure 1C, and all the fiber assays scheme should report the time of labeling with CldU and IdU.*

We included the duration of labeling for all the DNA fiber assay panels.

25. *Figure S1D (D) may refer to the wrong figure, please check.*

We updated the reference to figure S1D (now Supplementary Figure 1f).

26. *Figure 1E: the scheme representing the HU treatment should be expanded to show the incubation time.*

We included the duration of HU treatment in the schematic.

Reviewer #3 (Remarks to the Author):

In this study Rousseau et al., show that the Trim33 ubiquitin ligase limits replication-associated DNA damage induced by genotoxic agents and MYC oncogene. Mechanistically, authors show

that Trim33 regulates E2f4 transcription factor stability and restricts interactions of E2f4 with Recql helicase to control progression of DNA replication forks under replicative stress. Moreover, authors show that deletion of Trim33 triggers ectopic recruitment of E2F4 and Recql to chromatin, leading to rapid DNA replication, defective checkpoint signaling and DNA damage. Finally, by using an autochthonous model of liver cancer, authors show that Trim33 deficiency impairs survival under replicative stress and delays Myc-dependent liver tumorigenesis. The manuscript is interesting, and is overall properly done and well written, however, before publication can be recommended, several points need to be addressed in order to support the main conclusions of the work. With this aim, new experiments are suggested. Also, some points should be clarified or corrected.

We thank the reviewer for the interest and suggestions for experiments, which helped us to strengthen the manuscript.

Specific points that deserve additional attention are outlined below:

*- Although loss of Trim33 led to the deregulation of DNA replication and cell cycle genes, knockout of Trim33 did not significantly affect replication fork progression in unchallenged conditions. How are the cell cycle profile and DNA replication status of these cells? Moreover, the DNA damage present in these cells in unchallenged conditions should be assessed, since *gH2AX* levels were only measured upon HU treatment (Figure 6).*

We determined cell cycle distribution of unchallenged p19Nras Trim33-WT and Trim33-KO cells by FACS - the data are shown in Supplementary Figure 2b. The data for EdU incorporation in unchallenged Trim33-WT and Trim33-KO cells are provided in Figure 1f. We also compared the onset of DNA replication in these cells following release from G1 arrest, induced by serum deprivation or lovastatin treatment and found no significant differences - the data for serum deprivation are shown in Figure 1i.

We assessed pH2AX levels by immunofluorescence in unchallenged Trim33-WT and Trim33-KO cells, which shows a slightly reduced level in the knockout cells (Supplementary Figure 6a). The neutral comet assays for these cells under standard growth conditions are shown in Supplementary Figure 6e. Since the comet assays show no difference in the absence of stress, we believe that the small fold reduction in the pH2AX signal under standard growth conditions may reflect a compromised induction of checkpoint signaling during normal DNA replication in Trim33-KO cells.

- According to Figure 1, authors conclude that “depletion of Trim33 suppresses Myc-induced replicative stress and the accompanying activation of DDR signalling”. Strikingly, in the last section of the results authors state that “Loss of Trim33 exacerbates Myc-induced DNA damage”. These seem opposing messages and need clarification.

Indeed, we find lower levels of markers of replicative stress and DDR (pChk1, pS4/8-Rpa2, pKap1) in Trim33-KO cells compared to Trim33-WT cells in response to HU treatment (Figure 1g). This is consistent with accelerated DNA replication during release from HU and a reduced formation of 53bp1 bodies, which requires ATM signaling (Harrigan et al, 2011). In contrast, assessment of DNA damage by immunostaining with pH2AX antibodies, DSB capture and comet assays during

recovery from stress revealed accumulation of DNA damage. We believe that the simplest interpretation of these results is that Trim33-deficient cells fail to mount adequate stress response upon fork stalling, which compromises DNA repair.

- Related to the previous one, data shows that TRIM33 loss leads to increased gH2AX levels but decreased pKAP1 and pRPA DNA damage markers, as well as 53BP1 bodies. It would be necessary to discuss these conflicting data. Authors suggest that checkpoint signaling might be compromised upon Trim33 loss. Is anything known in this line?

As discussed in response to the previous point, we propose that rapid DNA replication and failure to mount an adequate signaling response to stalled forks in Trim33-KO cells leads to replication-associated DNA damage. As 53bp1 bodies are known to recruit regions, which are difficult to replicate, and require ATM signaling for assembly (Lukas et al, 2011; Harrigan et al, 2011), we surmise that compromised DDR signaling at stalled forks in Trim33-KO cells (indicated by lower pKap1 levels), impairs recognition of such underreplicated regions and DNA repair at these sites. We addressed this point in the Discussion section on page 10.

Previous studies showed that Trim33 is recruited to laser-induced DSBs in a Parp1-dependent manner and that loss of Trim33 sensitizes cells to genotoxic drugs (Kulkarni et al, 2013; McAvera et al, 2021). We cite these studies on page 9 in the Discussion section. Our data that Trim33 modulates recruitment of Recq1 by E2f4 are also consistent with previous observations that Recq1 can antagonize Parp1-dependent fork reversal (Berti et al, 2013).

- In Supplementary Figure 1D, DNA fiber assay is performed in U2OS cells with inducible expression of MYC, and authors conclude that TRIM33 loss rescues MYC induced fork slowing. However, this rescue is not very clear with knockout number 5. These analyses could benefit from additional fiber measurement, the measurement of 100 fibers in this kind of assay is usually preferred. Moreover, how were these U2OS knockout cell lines generated? In order to avoid long term effects of TRIM33 loss, I would recommend to perform acute depletion of TRIM33 in these assays. Moreover, if shRNAs were used, I would suggest to use "KD" instead of "KO" in Supplementary Figure 1C and D.

The data shown in Supplementary Figure 1D (now Supplementary Figure 1f) were obtained with U2OS cell lines generated using CRSPR-Cas9 genome editing. To ascertain that the impact of Trim33 knockouts on DNA replication are not due to long-term absence of Trim33, we transduced U2OS cells with lentiviruses expressing shRNAs against Trim33 and performed DNA fiber assays immediately after antibiotic selection and 24h induction of Myc with doxycycline. Similar to the phenotype of knockout cells, depletion of Trim33 rescued the slowing of replication forks after Myc induction, although the effects were weaker, likely due to the incomplete loss of the Trim33. These data are included as Supplementary Figure 1g,h.

We now provide data for at least 100 fibers per sample for all DNA fiber assays.

- Authors found that deletion of Trim33 stabilized E2F4 transcription factor, and their interaction might involve the DNA binding domain of E2F4. Considering that the DBD of E2F transcription factors is conserved among the different members of the E2F family of transcription factors, it would be interesting to know whether this function of Trim33 is specific for E2F4, or whether it regulates the stability of other E2F factors.

We compared the binding of Trim33 to E2f4, E2f1 and E2f3 in immunoprecipitation assays with benzonase-treated lysates of U2OS cells. These experiments showed that Trim33 selectively recruits E2f4 (Supplementary Figure 2g). We also assessed the stability of E2f1, E2f3 and E2f5 in p19Nras Trim33-WT and Trim33-KO cells using cycloheximide assays and did not find strong differences (Supplementary Figure 2f).

- Regarding the function of Trim33 in regulating E2F4 stability, the immunofluorescence images of E2F4 upon Trim33 deletion are not very clear. Quantification of the images would be helpful, as well as choosing new images that better represent the result. On the other hand, why are not E2F4 protein levels increased in steady state conditions upon Trim33 loss in Figure 2D? Moreover, since authors claim that DNA damage or MYC overexpression overrules Trim33 mediated E2F4 levels, are E2F4 levels increased under these conditions? Authors suggest that E2F4-Trim33 interaction is decreased upon HU, but data is not very convincing (Figure 5E). This is an important point, authors should try other approaches to make this point stronger.

We repeated the assay and quantified the immunofluorescence signal for E2f4 in Trim33-WT and Trim33-KO cells (Supplementary Figure 2e).

We found that E2f4 mRNA levels are lower in Trim33-KO cells compared to Trim33-WT cells (please see Figure 2 below), which reduces the apparent effect on steady state levels. However, because the turnover of E2f4 is strongly reduced by knockout of Trim33, the increase of E2f4 protein levels in Trim33-KO cells is still apparent in most experiments (Figure 2d, 4d, 3f, Supplementary Figure 2j).

Figure 2. Relative E2f4 mRNA levels in p19/Nras Trim33-WT and two Trim33-KO cell lines.

We found that treatment with different genotoxins in p19Nras cells or induction of Myc in U2OS cells lead to upregulation of E2f4 protein levels (Supplementary Figure 5d, f). We repeated the immunoprecipitation assays from HU-treated and unchallenged cells and provide a stronger

result in Figure 5f with quantification of three experiments. We also used PLA assays with E2f4 and Trim33 antibodies and obtain a comparable result (Figure 5g).

- Regarding E2F4 immunoprecipitations, there are several issues that should be considered. In Figure 2D, Trim33 band in E2F4 IP sample is very hard to see, even in the MG132 condition. Also, Co-IP analyses between E2F4 and Recql are not very clear. Performing these IP-s with exogenously expressed proteins or crosslinked samples would strengthen the data (as done in Figure 4B). Which band corresponds to E2F4 in Figure 2D? Regarding E2F4 mutants (Figure 2F), authors claim that E2F4-KR and E2F4-deltaC enhance the interaction with Trim33, however, Trim33 levels are also increased in the input samples. The decreased interaction between E2F4 and Trim33 upon HU treatment in Figure 5E, is hardly appreciated. Overall, IP data are not very strong and authors should improve them.

We repeated the E2f4-Trim33 immunoprecipitations and provide a stronger experiment for Figure 2d. To simplify the attribution of E2f4 bands, we now used a mouse anti-E2f4 antibody for IP followed by immunoblotting with a rabbit anti-E2f4 antibody, which results in a single band. We show the new data for immunoprecipitation of endogenous E2f4 followed by Recql immunoblots (Figure 3f). Following this suggestion, we immunoprecipitated endogenous Recql from lysate of formaldehyde-crosslinked Trim33-KO cells expressing HA-tagged E2f4 variants and present these data in Supplementary Figure 3e.

We agree that the input levels of Trim33 are higher for E2F4-KR and E2F4-deltaC than in the other samples. We therefore adjusted the description of this experiment (now Figure 2g) on page 4.

We repeated the IP experiments to assess the interaction of E2f4 and Trim33 in HU-treated cells (Figure 5f) and performed PLA assays to strengthen our conclusion (Figure 5g).

- According to Figure 2G, E2f4 depletion slowed down DNA replication forks during recovery from HU treatment in Trim33-KO cells, and expression of E2f4 variants that lack DNA binding domain or the C-terminal region, decelerated DNA replication and equalized fork progression in Trim33-WT and Trim33-KO cells (Figure 2H). Were experiments in Figure 2H performed without HU? If so, how do the authors explain the effect of these mutants considering that E2F4 depletion did not have any effect in unchallenged conditions? These data need better explanation and discussion.

The experiments shown in Figure 2g (now Figure 2j) were performed after release from a 4hr HU treatment, since deletion of Trim33 did not affect replication fork progression in unchallenged cells. We have now indicated this in the figure legend and in the main text. Notably, functional variants may act as dominant negative to fully inactivate endogenous protein, and have a stronger phenotype compared to siRNA-mediated depletion.

- Authors found that HU induced a robust recruitment of Recql to chromatin in Trim33-WT cells, which was especially prominent at E2f4-bound sites. It would be important to assess whether the

increase of Recql binding upon HU is the consequence of an increased E2F4 recruitment to these regions. Moreover, according to ChIP-Seq and RNA-Seq data, for most E2f4 binding sites, enhanced E2f4 recruitment upon Trim33 loss was not associated with altered gene expression. It would be interesting to assess if this is also the case upon HU treatment or MYC overexpression, at least in selected target genes. This may reinforce the idea that E2F4 has a non-transcriptional role in DNA damage conditions.

We used Cut&Run assays with E2f4 antibodies in Ctrl and HU-treated Trim33-WT cells and found a strong recruitment of E2f4 to chromatin upon HU (Figure 5h). E2f4 binding upon HU shows a good correlation with Recql recruitment (Figure 5i). Furthermore, siRNA-mediated depletion of E2f4 abolishes HU-induced recruitment of Recql at all of the examined transcription start sites (Supplementary Figure 5i). These results are in line with data obtained in Trim33-KO cells transfected with E2f4 siRNA or expressing E2f4 variants deficient in Recql or DNA binding, in which we find a strong reduction in Recql recruitment to chromatin (Figure 4e,f,g).

We compared gene expression in unchallenged and HU-treated cells using RNA-seq and found that E2f4 targets were not appreciably deregulated under these conditions (Supplementary Figure 5g). Less than 10% of loci that showed E2f4 recruitment, were in significantly deregulated genes and for these genes we found no correlation between E2f4 binding and gene expression (Supplementary Figure 5h).

- Regarding the analysis of transcription-replication conflicts in Figure 5G, H, authors should clarify why collisions for pS5-RNAPII were reduced upon HU treatment, whereas collisions for total RNAPII did not decrease upon HU in Trim33-WT cells. On the contrary, the interaction with PCNA of both, pS5-RNAPII and total RNAPII, were reduced upon Trim33 loss. Moreover, authors should check how Recql interaction with RNAPII and Mcm2 is affected upon HU.

Based on our data, we hypothesize that upon replicative stress, Recql is recruited transiently to E2f4-bound promoters (marked by pS5-RNAPII) in Trim33-WT cells, which underlies the selective effect on promoter-proximal TRCs. In contrast, complete loss of Trim33 leads to pervasive and much more broad association of Recql with chromatin, including gene bodies and transcription end sites (where total and pS2-RNAPII are enriched), which explains the effects on TRCs in the PLA assays with total RNAPII antibodies. We are addressing this interesting point experimentally, but it clearly requires significant time to finalize.

We performed immunoprecipitation assays with formaldehyde-crosslinked cells and found that HU induced robust association of Recql with RNAPII, Mcm2 and PCNA (Figure 5a).

- In Figure 6, authors used the E2f4-ΔC (which does not recruit Recql), mutant and observed decreased pHA2X levels in Trim33-KO cells following release from HU, and suggest that recruitment of Recql may underlie accumulation of DNA damage in the absence of Trim33. However, authors should consider that E2f4-ΔC mutant is not only deficient for Recql binding. These analyses might be performed together with Recql depletion, in order to strengthen the role of Recql in the increased DNA damage observed in Trim33 deficient cells. Moreover, in order to

strengthen that Trim33 loss promotes DNA damage and impairs cell growth via E2F4-Recql recruitment, colony survival assays should be performed upon E2F4 and Recql loss.

We depleted E2f4 and Recql using siRNA in Trim33-KO cells and determined the levels of DNA damage following release from HU treatment using immunofluorescence analysis with pH2AX antibodies and using neutral comet assays. Both experiments showed a significant reduction in DNA damage upon depletion of either E2f4 or Recql in Trim33-KO cells (Figure 6d,e). Expression of catalytically inactive Recql-K119A also diminished pH2AX levels in Trim33-KO cells after release from HU (Supplementary Figure 6d). Depletion of E2f4 and Recql also diminished DNA damage in Trim33-depleted cells with high Myc levels (Figure 7b,c), as did stable expression of Recql variant K119A (Supplementary Figure 7c).

To perform the colony assays we attempted to generate p19Nras cell lines with stable depletion of E2f4 or Recql using published lentiviral shRNA vectors but failed to obtain a strong knockdown. We are generating E2f4 and Recql knockout cell lines but the full validation of these cells will take much more time. We therefore assessed colony formation by Trim33-KO cells, expressing E2f4- Δ DB and Recql-K119A, which efficiently reduce pH2AX levels in Trim33-KO cells (Supplementary Figure 6c,d). Expression of either protein significantly improved survival of Trim33-KO cells after HU treatment (Supplementary Figure 6i), consistent with the effects on DNA damage.

- In Figure 4F, it would be important to include the fiber data of samples transfected with a scramble siRNA.

We included the data for scrambled siRNA-transfected cells (siCtrl) (now Figure 4h).

- If the E2f4-Recql interaction does not require the DNA-binding domain (Figure 3G), why do the authors suggest that E2f4 may facilitate Recq1 localization to chromatin?

Our data indicate that the C-terminal domain of E2f4 interacts with Recql, whereas the DNA-binding domain of the same E2f4 molecule binds its cognate sequences in the genome. This way, E2f4 can tether Recql to E2f4 binding sites in chromatin to facilitate association with proximally stalled replication forks.

- Molecular weights should be included in all the blots presented in the manuscript. This is especially important when analysing protein ubiquitylation.

We indicated the position of the MW markers for all immunoblots.

- For the RNASeq data, it is important to establish what criteria was used regarding the differential expression analysis.

We used a $p \text{ adj} < 0.01$ and no cutoff for $\log_2\text{FC}$ to define significantly deregulated genes in the analysis shown in Figure 1b. We added this information to the figure legend and the main text.

- *For all the cell treatments performed, it is important to include the treatment conditions in the figure legends.*

We added the description of the treatment conditions to the figure legends.

References

Benedict, B. *et al.* The RECQL helicase prevents replication fork collapse during replication stress. *Life Sci Alliance* **3** (2020).

Berti, M. *et al.* Human RECQ1 promotes restart of replication forks reversed by DNA topoisomerase I inhibition. *Nat Struct Mol Biol* **20**, 347-354 (2013).

Chappidi, N. *et al.* Fork Cleavage-Religation Cycle and Active Transcription Mediate Replication Restart after Fork Stalling at Co-transcriptional R-Loops. *Mol Cell* **77**, 528-541.e528 (2020).

Chen, C.-R., Kang, Y., Siegel, P. M. & Massagué, J. E2F4/5 and p107 as Smad cofactors linking the TGFbeta receptor to c-myc repression. *Cell* **110**, 19–32 (2002).

Harrigan, J. A. *et al.* Replication stress induces 53BP1-containing OPT domains in G1 cells. *J Cell Biol* **193**, 97-108 (2011).

Kulkarni, A. *et al.* Tripartite Motif-containing 33 (TRIM33) protein functions in the poly(ADP-ribose) polymerase (PARP)-dependent DNA damage response through interaction with Amplified in Liver Cancer 1 (ALC1) protein. *J Biol Chem* **288**, 32357-32369 (2013).

Lukas, C. *et al.* 53BP1 nuclear bodies form around DNA lesions generated by mitotic transmission of chromosomes under replication stress. *Nat Cell Biol* **13**, 243–253 (2011).

McAvera, R. M., Morgan, J.J., Mills, K.I., and Crawford, L.J. TRIM33 Loss in Multiple Myeloma Impairs the DNA Damage Response Resulting in Sensitivity to PARP and ATR Inhibitors. *blood*, 1570 (2021).

REVIEWERS' COMMENTS

Reviewer #1 (Remarks to the Author):

The authors have carefully revised the manuscript and have addressed all comments that I raised. In my view, this now is a very nice and important story that can be published.

Reviewer #2 (Remarks to the Author):

All my concerns were addressed adequately by the authors. I think this is a very thorough and insightful study, and I have no concerns left regarding its publication in Nature Communications.

Reviewer #3 (Remarks to the Author):

I believe the manuscript under consideration can be accepted for publication in the journal Nature Comm. after the authors properly addressed all my points. The new data are presented and discussed in the text in a convincing manner, and the improved manuscript is now very solid and thorough.

“Trim33 masks a non-transcriptional function of E2f4 in replication fork progression”

Rousseau et al.

Point-by-point response to reviewer comments

REVIEWERS' COMMENTS

Reviewer #1 (Remarks to the Author):

The authors have carefully revised the manuscript and have addressed all comments that I raised. In my view, this now is a very nice and important story that can be published.

We thank the reviewer for the helpful suggestions and appreciation of our work.

Reviewer #2 (Remarks to the Author):

All my concerns were addressed adequately by the authors. I think this is a very thorough and insightful study, and I have no concerns left regarding its publication in Nature Communications.

We thank the reviewer for many important suggestions and the positive comments.

Reviewer #3 (Remarks to the Author):

I believe the manuscript under consideration can be accepted for publication in the journal Nature Comm. after the authors properly addressed all my points. The new data are presented and discussed in the text in a convincing manner, and the improved manuscript is now very solid and thorough.

We are grateful to the reviewer for the constructive suggestions for revisions and the positive feedback.